# MIND: Market Interpretation DSL for Unified Market Design and Simulation

## Abstract

Market mechanisms such as auctions and matchings coordinate supply and demand at scale, yet their implementations remain locked in rigid procedural code that hinders iteration and auditing. We introduce the Market Interpretation DSL (MIND), a typed language and toolchain for declarative market specification to achieve unified market design and simulation. MIND comprises (i) a core grammar with a phased Intermediate Representation (IR) and economic safety checks, (ii) a natural language assistant that translates descriptions into DSL with automated diagnostics and safe rewrites, and (iii) rule-based simulation and convex optimization backends. Using synthetic specifications generated across 87 domains with held-out validation, our fine-tuned Llama-3-8B assistant achieves 96.33% semantic correctness, measured as IR equivalence to gold programs, surpassing few-shot GPT-4o at 81.11%. Across second-price auctions, multi-stage auctions, and matching markets, MIND reduces specification complexity by approximately 79% in lines of code compared to Python implementations. In a within-subjects study with 17 participants, mechanism modifications were completed 4 to 10 times faster using MIND. Code, dataset, and models will be released upon acceptance.

## 1 Introduction

Market mechanisms such as auctions and matching markets form the backbone of modern economics, digital platforms, and decentralized systems. They coordinate supply and demand, reduce transaction costs, and enable efficient allocation of scarce resources (Milgrom, 2004; Roth, 2018; Milgrom, 2021). Despite this centrality, practical modeling and implementation remain cumbersome. Most platforms and simulators still hard-code allocation rules, matching logic, and pricing routines into procedural code, creating a lossy translation from policy to code (Calheiros et al., 2011; Byrd et al., 2019). This limits experimentation and complicates verification of market properties.

Beyond performance, platform operators must ensure transparent rules and reproducible outcomes for regulatory compliance, requiring a chain from human-readable policies to executable logic with audit trails. Decentralized trading systems raise the bar further: mechanism logic executes on-chain and requires formal checks for correctness and economic safety (d'Eon et al., 2024; Bouaicha et al., 2025). The core limitation is the absence of a unified interface that bridges conceptual specifications to deterministic implementations with support for governance, testing, and audit.

Recent LLM-based approaches to mechanism automation yield non-deterministic outputs and brittle patches, making debugging difficult where fairness and correctness are paramount. They also struggle to bridge underspecified natural language and verbose implementations, frequently omitting crucial details such as reserve prices, tie-breaking rules, or budget constraints. Moreover, the primary users are economists and policy analysts who possess domain expertise but typically lack programming skills. While GUI-based tools exist, they cannot express conditional constraints or multi-stage interactions, reducing mechanisms to rigid templates.

We advocate a domain-specific language paired with natural language translation that separates authoring, validation, and execution. A compact, typed DSL makes specifications legible, enables static checks for economic consistency, and supports deterministic compilation to multiple backends for cross-validation. This creates a governance surface where specifications carry provenance, version identifiers, and audit traces, while validators enforce safety gates before deployment. The system serves both audiences: domain experts use natural language to generate initial specifications,

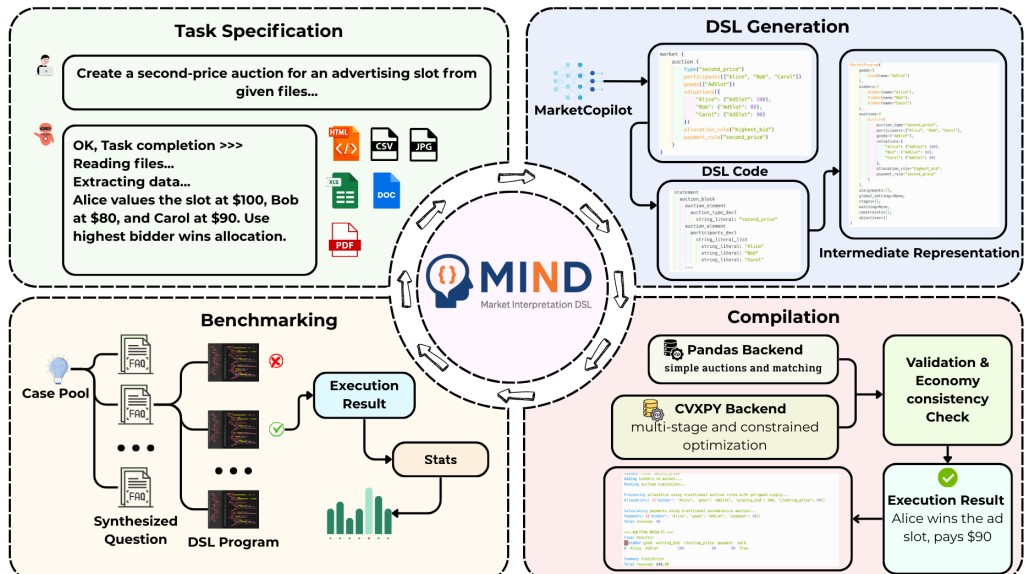

Figure 1: The whole workflow of our system. First, the Completeness Agent helps users with complete task descriptions. Then, the AI Copilot (MarketCopilot) translates the refined description into MIND programs. Finally, the compiler executes the programs with appropriate backends to produce the final results.

while power users can directly edit DSL programs. In MIND, semantic correctness is evaluated as Intermediate Representation (IR) equivalence to reference programs, and specification complexity is measured in AST nodes.

In this paper, we present MIND (Market Interpretation DSL), a comprehensive and extensible language and toolchain for specifying, validating, and executing market mechanisms. MIND defines a grammar of objects, actions, and types, compiled into a phased IR which is automatically validated and compiled into executable simulations, thereby decoupling specification from execution. The system includes a natural language assistant that translates descriptions into DSL programs, provides diagnostics, and applies safe rewrites (Zhang et al., 2023). Two backends support rule-based simulation and convex optimization. Each specification is a versioned artifact with machine-checkable metadata.

We evaluate MIND along three axes. First, a workflow study shows a 79% reduction in specification complexity versus Python. Second, a within-subjects user study ($N = 17$) confirms that practitioners modify mechanisms 4–10× faster in MIND than in a Python baseline. Third, we demonstrate generalization by encoding 8 mechanisms from recent ACM EC/SIGecom literature without changing the core grammar. Additionally, our fine-tuned Llama-3-8B Copilot achieves 96.33% semantic correctness on NL→DSL translation.

Our contributions are threefold:

- We introduce MIND, a domain-specific language with formal grammar and a phased IR that bridges natural language to executable simulations while creating auditable specifications.
- We develop an execution framework with two backends that support rule-based simulation and convex optimization from unified specifications, enabling deterministic cross-validation for compliance.
- We build a natural language to DSL translation system, achieving 96.33% semantic correctness across 87 domains, serving both non-programmers and power users.

## 2 RELATED WORK

**Market mechanism DSLs.** Prior mechanism modeling in economics has largely relied on general-purpose programming or specialized simulators, making specification and validation cumbersome. Recent work explores domain-specific languages to capture auction rules, negotiation

games, or fair division protocols in symbolic form (Hoseindoost et al., 2024; De Jonge & Zhang, 2021; Bertram et al., 2023). CoorERE (Hoseindoost et al., 2024) provides an executable DSL for auction-based coordination in crisis response, reducing development effort by nearly half, but it addresses single-item auctions without cross-mechanism support. GDL has been repurposed as a unifying description language for negotiation domains (De Jonge & Zhang, 2021), enabling generic solvers, but it lacks intermediate representations with economic validation. Slice (Bertram et al., 2023) defines a DSL for fair division protocols with automated envy-freeness verification, yet remains limited to division problems without auction or matching support. These DSLs improve mechanism specification but are scoped to individual subdomains and do not provide staged validation, two execution backends, or governance artifacts that MIND includes.

**LLMs in mechanism design.** The rise of large language models has motivated new approaches to automating specification and simulation. Recent studies use LLMs to generate valuations, bidding policies, and to propose new auction formats (Duetting et al., 2024; Sun et al., 2024; Dubey et al., 2024b; Shah et al., 2025). LaMP-Val (Sun et al., 2024) uses GPT-4 to infer personalized valuations from text and fine-tunes smaller models as strategic agents. Dubey et al. (Dubey et al., 2024b) and Duetting et al. (Duetting et al., 2024) examine auctions where advertisers bid for influence over LLM outputs, proposing incentive-compatible rules for token-level allocation. Shah et al. (Shah et al., 2025) show GPT-4 agents can reproduce human-like bidding behaviors, suggesting LLMs can serve as synthetic participants. These approaches demonstrate potential for synthesis and simulation but operate without typed specifications, deterministic compilation, or audit trails. They generate code directly without an intermediate representation, making systematic verification and governance difficult. They often lack empirical validation of generated mechanisms against ground truth specifications.

**Unified frameworks and positioning.** Prior DSLs achieve domain-specific expressiveness and LLM approaches enable automation, yet the literature remains fragmented: CoorERE focuses on crisis response, Slice on fair division, and LLM methods typically lack formal specifications. Technical barriers to unification include incompatible type systems across auction and matching domains, the absence of staged validation for economic properties, and limited support for governance requirements such as provenance tracking and policy diffs. MIND addresses these gaps through a unified grammar spanning auctions, matchings, and exchanges; an intermediate representation with three-stage validation (parsing, typing, economic consistency); two execution backends for simulation and optimization that scale to thousands of participants; natural language translation achieving 96.33% semantic correctness on 87 domains; and governance artifacts including versioning, validator reports, and audit logs. This combination links formal specification, property verification, and agent-based evaluation in a single reproducible workflow. Our evaluation shows it reduces specification complexity by 79% while maintaining semantic accuracy comparable to hand-written implementations.

**Verification and Guardrail Systems.** Beyond synthesis, ensuring safety requires rigorous verification. Formal verification engines like Imandra (Passmore et al., 2020) and certified auction frameworks (Caminati et al., 2015) use theorem proving to guarantee properties like incentive compatibility. In industry, frameworks like AWS Bedrock Guardrails enforce output safety policies. MIND complements these systems rather than replacing them: by producing a typed, distinct Intermediate Representation (IR) rather than opaque Python scripts, MIND provides the necessary structured input that these advanced verification engines require to perform mathematical proofs and policy enforcement.

## 3 METHOD

As illustrated in Figure 1, our system provides an end-to-end pipeline for generating, validating, and simulating MIND, starting from a user specification. The architecture is composed of several key parts: (1) a symbolic DSL for formal representation (Shi et al., 2024; Borum & Seidl, 2022), (2) an Intermediate Representation (IR) with a robust validation system, (3) a two-backend framework for code generation, and (4) an AI-powered toolchain including a dataset generation pipeline, a completeness agent, and a fine-tuned AI Copilot (referred to as MarketCopilot) for NL→DSL translation.

## 3.1 MARKET INTERPRETATION DSL

The foundation of our system is Market Interpretation DSL (MIND), a formal language designed for the specification of market rules. The language's grammar is built on a clear separation of core concepts: (i) **Objects** are entities that constitute a market, such as auction, participants, goods, and matching; (ii) **Actions** are operations that define the market's behavior, such as specifying the auction type, defining valuations, or setting constraints; (iii) **Types** are specific variants of objects and actions, like `type("second_price")` or `type("first_price")`. Some DSL examples are shown in Figure 2.

**Scope of Expressiveness** From the current grammar and IR, MIND is designed to support: (1) single- and multi-stage sealed-bid auctions with standard allocation and payment rules; (2) compatibility-constrained matching markets (e.g., stable matching (Gale & Shapley, 1962)) with explicit compatibility graphs; and (3) mechanisms with convex objectives and linear constraints. We explicitly detail the supported features versus out-of-scope capabilities in Appendix Table 4. Note that we do not currently support fully general combinatorial bidding languages (e.g., arbitrary XOR bids without constraints (Nisan, 2006)) or iterative open-cry auctions (e.g., ascending clock auctions (Ausubel, 2004)); these are planned for future extensions.

## 3.2 INTERMEDIATE REPRESENTATION (IR) AND VALIDATION

A challenge in designing a language with multiple execution targets is preventing our language parser from getting entangled with the specific details of every execution backend (Pandas, CVXPY, etc.). This creates a brittle, unscalable system where adding a new backend or modifying the DSL syntax would require cascading changes across the entire codebase.

To solve this problem, we introduce an Intermediate Representation (IR) (Lattner et al., 2021) as a critical abstraction layer. The IR serves as the **single source of truth** for mechanism semantics and governance, not just an intermediate parsing artifact. It is a typed abstract syntax tree (AST) over market constructs (e.g., `AuctionNode`, `ConstraintNode`). The parser translates DSL source into IR only; code generators read IR only. This separation ensures modularity and maintainability. Crucially, IR nodes, validator report identifiers, and compile artifacts are logged with each execution, so any result can be replayed with the exact spec and validator configuration.

To ensure that any market specified in the DSL is not just syntactically correct, but also semantically and economically sound, the IR undergoes a rigorous validation process before generating code. This process consists of three phases: parsing, typing, and economic consistency. Three main validators run in order on the IR:

1. **CoreMarketValidator:** Performs fundamental checks, ensuring names are unique, references are valid, valuations align with participants, and auction rules are recognized.

2. **StageAndMatchingValidator:** If the design uses stages or matching, this validator runs to perform checks on global settings and validate the structure of these advanced components.

3. **AdvancedOptimizationValidator:** If the design includes constraints or objectives, this validator checks that their types are recognized and parameters are valid (e.g., non-negative budgets).

Each validator consumes an IR snapshot and emits a `ValidationReport` with typed findings (`error`, `warning`, `autofixable`). The Autofixer applies only safe rewrites; if a required rule cannot be inferred, it emits a blocking error rather than altering semantics. We persist the spec hash, validator report identifier, and compile artifact path with the run logs to enable exact reconstruction during audit. All experiments log these identifiers, allowing any reported result to be traced to its exact specification and validator state.

## 3.3 TWO-BACKEND CODE GENERATION

While theoretically a monolithic engine could handle all designs, we treat backend selection as an **engineering trade-off** between performance and solver generality. Simple single-shot auctions and matching markets benefit from fast, vectorized table-driven simulation, whereas constrained or combinatorial designs require solver-grade convex optimization. To handle both without exposing backend complexity to users, we compile the same backend-independent IR—the single source of truth—into different execution targets via `MarketCompiler`.

```
AUCTION DSL SAMPLE

market {
    auction {
        type("second_price")
        participants(["Alice", "Bob", "Carol"])
        goods(["AdSlot"])
        valuations({
            "Alice": {"AdSlot": 100},
            "Bob": {"AdSlot": 80},
            "Carol": {"AdSlot": 90}
        })
        allocation_rule("highest_bid")
        payment_rule("second_price")
    }
}
```

```
BIPARTITE MATCHING DSL SAMPLE

market {
    matching {
        type("bipartite")
        participants(["Alice", "Bob", "Carol", "General", "Cardiac", "Neuro"])
        compatibility_graph({
            "Alice": ["General", "Cardiac"],
            "Bob": ["Cardiac", "Neuro"],
            "Carol": ["General", "Neuro"],
            "General": ["Alice", "Carol"],
            "Cardiac": ["Alice", "Bob"],
            "Neuro": ["Bob", "Carol"]
        })
        matching_rule("stable_matching")
    }
}
```

Figure 2: Two MIND specifications. Left: second-price auction where `type` specifies auction type, `participants` lists bidders, `goods` declares the item, `valuations` gives each bidder's valuation, `allocation_rule()` assigns to the highest bidder, and `payment_rule()` charges the second-highest bid. Right: simple matching market specification.

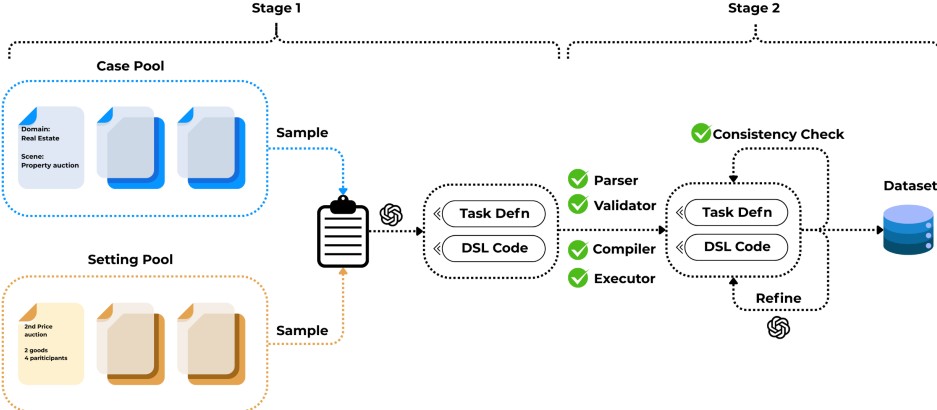

Figure 3: Market Mechanism Dataset pipeline with three phases: data generation, augmentation, and filtering.

**Backend routing** `MarketCompiler` selects a backend via a deterministic function of IR features: designs without explicit objectives or global constraints are routed to the **Pandas/NetworkX (Simulation)** backend; those declaring objectives or linear constraints are routed to the **CVXPY (Optimization)** backend. Crucially, all backends must implement identical IR semantics; observable outcomes (allocations, payments) must agree for any given specification, enabling **cross-backend consistency checks** for governance.

- **Pandas Backend:** A lightweight, simulation-based backend for standard auctions and matching markets, optimized for speed and simplicity.
- **CVXPY Backend:** An optimization-based backend using the CVXPY library (Diamond & Boyd, 2016), automatically selected for scenarios involving constraints (e.g., budget balance) or objectives (e.g., maximizing revenue).

Backends implement the same IR semantics; observable outcomes (allocations, payments, feasibility flags) must agree for identical IR inputs. This two-backend approach lets users scale from simple simulations to constrained optimizations without changing the DSL specification, while preserving consistent semantics across backends.

## 3.4 DATASET GENERATION

Supervised fine-tuning of a specialized Copilot requires a large-scale, high-quality dataset of (Natural Language Description, DSL) pairs. To the best of our knowledge, there is no such dataset for the task of translating natural language specifications into a formal DSL for market mechanisms. To

address this, we developed an automated pipeline (Ratner et al., 2017; Northcutt et al., 2021) to use LLMs to generate synthetic data as illustrated in Figure 3. The process begins by programmatically generating diverse prompts for a generator LLM (GPT-4o (Hurst et al., 2024)). We predefine over 900 possible market use cases within 87 domains. For each use case, we randomly sample settings to generate prompts with the formal DSL grammar and in-context examples.

To ensure correctness, every DSL program undergoes a rigorous 4-stage validation pipeline: (1) grammar parsing, (2) three-phase IR validation (structure, typing, economic consistency), (3) compilation to both backends, and (4) execution without runtime errors.

After guaranteeing code correctness, we refine the corresponding natural language descriptions. Each validated DSL program is passed to an LLM to generate a more detailed and complete description. As a quality control step, another verifier LLM performs a description-DSL consistency check, confirming semantic alignment between the enhanced natural language description and the DSL code. Finally, beyond these automated checks, we conducted a **manual inspection** of the candidate pool to filter out any remaining low-quality or redundant samples. Only pairs passing this multi-stage verification are included in the dataset.

### 3.4.1 HUMAN AUDIT PROTOCOL

To ensure the quality of our automated pipeline, we perform a manual human audit. We drew a simple random sample of 100 (description, DSL) pairs from the final, post-filter dataset after the 4-stage validation and description-DSL consistency check, stratified by domain and mechanism type. An expert evaluator (one of the authors) assessed each pair on: (i) syntactic correctness (DSL parses under the grammar), (ii) semantic alignment (IR equivalence of the compiled DSL to the behavior described), and (iii) functional executability (successful backend compilation and execution). The evaluator was blinded to the specific pipeline metadata during assessment to minimize bias. A pair passes only if all three criteria are satisfied. The audit confirms a very high accuracy rate under these criteria.

### 3.5 COMPLETENESS AGENT

To ensure the generated DSL program fully aligns with users' expectations, we need descriptions with sufficient detail. Since users rarely provide complete descriptions initially, we developed a Completeness Agent (Yao et al., 2023; Shinn et al., 2023) as a pre-processor to help users provide enough information for the MarketCopilot.

The agent operates through a multi-node scheme defining key elements to capture. For each node, it extracts information from users' prompts and populates the scheme with required and optional fields. If all required fields are fulfilled, the node completes and the agent proceeds. Otherwise, it requests missing critical information. After passing all nodes, the agent outputs a *completion schema* and a *complete task specification* that are passed to the MarketCopilot. The completion schema maps directly to the MarketCopilot input fields (participants, goods, constraints, objectives), ensuring the prompt is structurally complete before translation. This significantly increases the likelihood of generating a valid and executable DSL program on the first attempt.

### 3.6 MARKETCOPILOT FINETUNING

The core of our natural language interface is the **AI Copilot (MarketCopilot), an AI assistant fine-tuned for NL-to-DSL translation. It is distinct from the Completeness Agent (which only fills missing fields) and the DSL framework itself.** We used a Llama-3-8B-Instruct model (Dubey et al., 2024a) as our base and applied LoRA (Hu et al., 2022) with rank $r = 32$ for efficient training. The model was trained on our curated dataset using a standard supervised fine-tuning (SFT) objective to maximize the conditional probability of generating the ground-truth DSL program given the natural language description. The training loss is:

$$\mathcal{L}(\theta) = - \sum_{(X_i, Y_i) \in \mathcal{D}} \log P(Y_i | X_i; \theta) \tag{1}$$

Here, $\mathcal{D}$ is the set of (description $X_i$, DSL sequence $Y_i$) pairs; $Y_i$ is tokenized as a left-to-right sequence for teacher forcing under the SFT objective in Eq. 1.

### 3.7 Pluggable Frontend Interfaces

We view natural-language and graphical user interfaces (GUIs) as complementary rather than competing approaches. The MIND architecture is designed with a **pluggable front-end**: the core DSL and IR remain invariant regardless of the input modality. To validate this, we developed a **prototype web GUI** where users input natural language descriptions, and the Completeness Agent runs interactively to ask follow-up questions and populate a structured schema before passing it to the MarketCopilot. This demonstrates that MIND supports diverse workflows—whether pure natural language, form-based, or hybrid—all targeting the same unified, verifiable DSL and IR.

## 4 Experiments

To validate our system, we designed two primary experiments. First, we evaluate the Market Interpretation DSL itself by comparing its workflow and expressiveness against standard procedural programming approaches for market design. Second, we quantitatively evaluate the performance of our fine-tuned AI Copilot in translating natural language specifications into valid and semantically correct DSL code.

### 4.1 DSL Workflow and User Study Evaluation

The primary motivation for creating a Domain-Specific Language is to accelerate development, and reduce errors. We evaluate MIND's impact on market design workflows at two levels: (i) a code-level comparison of hand-written implementations (Table 1) to measure specification complexity; and (ii) a within-subjects user study with 17 participants comparing MIND against a Python baseline to measure modification efficiency.

#### 4.1.1 Methodology and Benchmark Tasks

**Code-Level Benchmarks (Table 1).** To quantify specification complexity, we implemented three representative market mechanisms: (1) A standard second-price auction; (2) A multi-stage auction with reserves; and (3) A compatibility matching market. We implemented each using MIND, standard Python, and **AnyLogic** (Borshchev & Filippov, 2004). We evaluated these approaches based on three established software engineering criteria. **Specification Complexity** refers to the effort required to define the mechanism, measured by the number of distinct modeling steps and Source Lines of Code (LoC) (Molnar & Motogna, 2020). **Readability & Verifiability** describes how easily the implementation can be audited against its theoretical design, a crucial aspect of model correctness (Alawad et al., 2019). Finally, **Flexibility** measures the effort required to modify an existing mechanism (e.g., changing a pricing rule), which is a key indicator of software maintainability (Ardito et al., 2020).

**User Study Methodology.** We recruited 17 participants with varyinh levels of expertise. The majority had >2 years of Python experience. The study followed a within-subjects design where each participant performed modification tasks on the three benchmark mechanisms in both MIND and Python environments. The tasks required modifying pricing rules, adding re-auction stages, and adjusting compatibility constraints. We recorded Completion Time (self-reported minutes) and Correctness (verified offline). Detailed protocol and participant demographics are provided in **Appendix I**.

#### 4.1.2 Results and Analysis

**Specification Complexity** (Table 1). The results of our code-level comparison demonstrate the significant advantages of the DSL-based approach. As summarized in Table 1, `MIND` consistently requires the least specification effort. Across all three tasks, MIND reduces specification complexity by approximately 79% in lines of code compared to Python implementations. Its declarative syntax allows designers to focus on economic rules rather than procedural control flow. In contrast, the Python approach requires significant boilerplate code and embeds the core mechanism logic within procedural control flow, making it difficult to verify and modify. While powerful, AnyLogic introduced a high degree of complexity and a steep learning curve, making it not that suits for the rapid prototyping of mechanism rules, which is a primary goal of our system.

**User Study Results.** The results of our user study indicate a significant workflow advantage for the DSL. Participants completed the mechanism modifications consistently faster using MIND com-

Table 1: A comparison of implementation workflows for common market design tasks. Our declarative approach, MIND, consistently requires the least specification effort, offers the highest clarity for verification, and provides the greatest flexibility for experimentation.

| Task | Approach | Specification Complexity | Readability & Verifiability | Flexibility (Effort to Modify) |
|------|----------|--------------------------|-----------------------------|--------------------------------|
| Second-Price Auction | MIND | ~10 lines of DSL | High: Declarative economic syntax. | Trivial: Change a single keyword. |
| | Python | ~40-60 lines of code | Low: Core logic is embedded in code. | Moderate: Requires rewriting functions. |
| | AnyLogic | ~15-20 graphical steps | Medium: Logic is distributed across agents. | High: Requires reconfiguration. |
| Multi-Stage Auction | MIND | ~20-25 lines of DSL | High: Staging logic is explicit and easy to follow. | Trivial: Modify a self-contained stage block. |
| | Python | ~100-120 lines of code | Low: State management is complex and error-prone. | High: Requires significant refactoring of the main control flow. |
| | AnyLogic | ~25-35 graphical steps | Low: Managing agent state across stages is hard. | High: Requires a full redesign of the simulation flowchart. |
| Matching Market | MIND | ~15 lines of DSL | High: The compatibility graph is a direct input. | Low: Change the data directly. |
| | Python (w/ NetworkX) | ~60-80 lines of code | Medium: Requires graph library expertise to understand. | Moderate: Requires implementing a different matching algorithm. |
| | AnyLogic | ~35-45 graphical steps | Medium: Requires defining custom agent interaction rules. | High: Requires creating new agent protocols. |

pared to Python (with median speed-up between 4-10x). Detailed participant demographics, task breakdowns, and full statistical reports are provided in **Appendix I**.

## 4.2 COPILOT GENERATION EVALUATION

This experiment evaluates the ability of our AI Copilot (MarketCopilot) to automatically generate high-quality Market Interpretation DSL code from natural language descriptions.

### 4.2.1 EXPERIMENTAL SETUP

We evaluated our fine-tuned MarketCopilot (Llama-3-8B + LoRA) against several baseline models on a held-out test set of 323 examples across 87 domains, ensuring no overlap with training data beyond a 0.85 cosine similarity threshold computed on TF-IDF representations. We additionally exclude near-duplicates by AST hash to prevent leakage. The training set consisted of 11,000 examples, with 10% used for validation during hyperparameter tuning.

To provide a robust comparison, baseline models were evaluated in a few-shot setting with the formal DSL grammar specification and 4 examples of (NL, DSL) pairs along with the task description. In contrast, our MarketCopilot operates zero-shot, taking only the natural language task description as input.

### 4.2.2 EVALUATION PIPELINE AND METRICS

We employed a rigorous multi-stage pipeline to assess correctness:

1. **Grammar Validation:** Each output is parsed using our Lark EBNF grammar. We measure Parse Success Rate as the percentage of syntactically valid programs.

2. **Semantic Validation:** Syntactically correct programs undergo three checks:
   - *Validator check:* Tests logical consistency using the three-phase validation (parsing, typing, economic consistency)
   - *Compiler check:* Verifies code generation to the two execution backends

Table 2: Performance of the AI Copilot against baseline LLMs on the NL-to-DSL generation task (323 test cases). **Our fine-tuned model outperforms significantly larger proprietary models**, demonstrating that strict grammar compliance requires domain adaptation rather than just reasoning power.

| Model | Parse Success (%) | Val + Comp Success (%) | IR Equivalence (%) |
|---|---|---|---|
| *Proprietary SOTA Models (Few-shot)* | | | |
| GPT-4o-mini | 97.52 | 89.47 | 76.78 |
| GPT-4o (Hurst et al., 2024) | 98.45 | 95.98 | 81.11 |
| Gemini 2.5 Pro (Reasoning) | **99.69** | 99.07 | 77.71 |
| GPT-5 (Reasoning) | 96.90 | 92.57 | 72.12 |
| *Open-Source Models* | | | |
| Llama-3-8B | 80.76 | 73.71 | 60.89 |
| Qwen3-Coder-30B(Yang et al., 2025) | 95.51 | 94.55 | 81.73 |
| **Our Method (Llama-3 + LoRA)** | **100.00** | **100.00** | **96.33** |

- *IR Semantic Equivalence:* We define this as graph isomorphism between the generated and reference IR ASTs, modulo variable renaming, commutative ordering, and alias normalization

3. **Execution Validation:** Programs are executed on 323 test scenarios to verify they produce correct market outcomes (allocations, payments, feasibility). Scenarios mirror the functional spec used in the workflow study.

Our primary metric is **End-to-End Correctness**: the percentage of generations that pass all validation stages and are semantically equivalent to the reference solution. We define this strictly: a generation is correct **if and only if** it: (1) parses successfully, (2) passes all three validation stages, (3) compiles successfully to both backends, (4) achieves IR semantic equivalence to the ground truth, and (5) passes execution checks.

### 4.2.3 RESULTS AND DISCUSSION

Table 2 presents the performance metrics. Our fine-tuned model achieves the highest scores across all metrics, with **96.33% end-to-end correctness**, significantly outperforming the best general-purpose baseline (GPT-4o at 81.11%).

Crucially, our experiments reveal that stronger reasoning models do not necessarily yield better DSL specifications. The reasoning-focused **GPT-5** model achieves only 72.12% correctness, performing worse than GPT-4o. Error analysis indicates that powerful reasoning models tend to "over-think" the task: they frequently hallucinate plausible but unsupported keywords or attempt to restructure the mechanism logic in ways that violate the DSL schema. This highlights a key trade-off: while large models excel at unstructured reasoning, domain-specific fine-tuning is essential for strict adherence to formal grammars.

It is also worth noting that MIND is **frontend-agnostic**. While our current implementation uses a fine-tuned 8B model for efficiency and controllability, the underlying IR, validator, and compiler pipeline can serve as a robust guardrail for any future foundation model.

### 4.3 GENERALIZATION TO ACADEMIC MECHANISMS

To assess MIND's capability to generalize beyond standard textbook examples, we implemented 8 mechanisms from recent ACM EC and SIGecom papers (2020–2024). These include complex designs such as dynamic auction throttling, randomized clock auctions, and stochastic ridesharing matching.

**Results.** All 8 mechanisms were successfully encoded in MIND, passed the 4-stage validation pipeline, and executed on the appropriate backend (simulation or optimization). Crucially, none required changes to the core MIND grammar; specific logic was handled by adding rule names (e.g., `"dynamic_price_floor"`) as library entries or composing existing blocks (e.g., `dynamic + cost_function`). We provide the detailed mapping of these papers to MIND constructs in Appendix H.

# 5 ABLATION STUDIES

To quantify the impact of our data curation process, we conduct ablation studies on progressively less-filtered versions of our training dataset: (1) **Parse-Only**: filtered only for syntactic correctness; (2) **No Execute Check**: filtered through compilation (Parse → Validator → Compiler) but without execution-time validation; (3) **No LLM Check**: passes full 4-stage validation (Parse → Validator → Compiler → Execute) but lacks the description-DSL consistency verification. We train separate MarketCopilot models on each dataset variant using identical architectures, training budgets, and hyperparameters.

Table 3: Ablation study results demonstrating the impact of each data curation stage. All models trained with identical architectures and budgets.

| Model | Parse Success (%) | Validation + Compilation Success (%) | IR Equivalence (%) | ΔIR vs Previous (pp) |
|---|---|---|---|---|
| Parse-Only | 98.33 | 81.33 | 66.33 | – |
| w/o Execute Check | 98.67 | 92.33 | 68.67 | +2.34 |
| w/o LLM Check | 100.00 | 99.33 | 72.33 | +3.66 |
| **Full Pipeline** | **100.00** | **100.00** | **96.33** | **+24.00** |

Table 3 demonstrates that each curation stage contributes significantly to final performance. While parse success remains uniformly high (>98%) across all variants—indicating that learning basic DSL syntax is straightforward—the gaps emerge in semantic correctness. Validation and compilation success improves from 81.33% to 100% as filtering stages are added, with the execution check contributing 7.00 percentage points and validator checks contributing 11.00 percentage points from the Parse-Only baseline.

Most critically, IR equivalence shows dramatic improvement: from 66.33% (Parse-Only) to 96.33% (Full Pipeline), a total gain of 30.00 percentage points. This **30-point gain** is driven by the IR-level validation and description–DSL alignment, not by a larger LLM. The description-DSL consistency check alone contributes 24.00 percentage points (72.33% to 96.33%), highlighting that alignment between natural language and formal specifications is crucial for semantic correctness. Without this final verification, models generate syntactically valid but semantically incorrect programs—they learn surface patterns rather than the underlying mapping between economic concepts and their formal representations.

These results validate our design choice to prioritize data quality over quantity. Training on carefully curated examples produces models that understand the semantic correspondence between natural language descriptions and market mechanisms, rather than merely mimicking syntactic patterns.

# 6 CONCLUSION AND FUTURE WORK

We present MIND, a symbolic language and toolchain that bridges economic design and executable implementation. Through a typed IR, phased validation, and dual-backend execution, MIND reduces specification complexity by 79% and enables an AI Copilot to achieve 96.33% semantic correctness. A within-subjects user study confirms that practitioners modify mechanisms 4–10× faster in MIND than in Python. We further demonstrate generalization by encoding 8 mechanisms from recent ACM EC/SIGecom literature. Designed to be front-end agnostic, MIND allows stronger reasoning models to plug into its governance layer—comprising spec hashes, validator reports, and audit logs—to ensure traceability and compliance.

Future work focuses on expanding scope and integration. We plan to support combinatorial bidding languages, iterative auctions, and stochastic Bayesian games, while scaling to millions of participants. Crucially, we aim to bridge MIND with external ecosystems: this includes handling schema drift in messy real-world data and integrating with formal verification engines (Passmore et al., 2020) to mathematically prove economic properties. With these efforts, MIND provides a practical foundation for exploring, auditing, and deploying market designs.

REPRODUCIBILITY STATEMENT

Our work is committed to the principles of open and reproducible research. To this end, all code, datasets, and experimental configurations will be made publicly available upon acceptance of this paper.

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

## USE OF LLMs in our work

We used large language models (LLMs) in four ways: (i) manuscript polishing—to improve grammar, clarity, and flow without altering substantive claims; (ii) literature triage—to surface potentially relevant papers; (iii) data creation—to synthesize a portion of our NL→DSL pairs (see Sec. 3.4); and (iv) *prompt design*—to iterate on task instructions and few-shot exemplars. All LLM outputs affecting results were reviewed by authors for accuracy, and dataset items were validated with our parser/validator pipeline and spot-audited by humans.

## A   MARKET INTERPRETATION DSL COMPONENTS

The core building blocks and their overall structure are:

```
market {
  global_settings { ... }
  auction { ... } // repeatable
  stage(name="...") { ... } // optional, repeatable
  matching { ... } // optional
  constraints { ... } // optional
  objectives { ... } // optional
  dynamic { ... } // optional
}
```

**Block Descriptions**

- **market**: Top-level container for a complete market specification.
- **global_settings** (optional): Global parameters (e.g., units, supply, defaults).
- **auction** (repeatable): Auction mechanism definition (type, participants, goods, valuations, rules, distributions).
- **stage** (optional, repeatable): Multi-stage orchestration with re-auctioning and discounting logic.
- **matching** (optional): Matching market (type, participants, compatibility, rule).
- **constraints** (optional): Feasibility/policy conditions; simple or parameterized forms.
- **objectives** (optional): Optimization goals used by solver backends.
- **dynamic** (optional): Settings for time-varying parameters, multi-period loops, and discounting.

### A.1   SUPPORTED MECHANISM SCOPE

Table 4: MIND Scope of Expressiveness. We categorize features into currently supported (In-Scope) and those reserved for future work (Out-of-Scope).

| Feature Category | Supported (In-Scope) | Out-of-Scope |
|---|---|---|
| **Auction Formats** | • **Standard**: First-Price, Second-Price, Uniform-Price, Pay-as-Bid.
• **Combinatorial**: Additive valuations with logical constraints (AND/OR, XOR via constraints). | • **Complex Combinatorial**: General XOR bidding languages without constraints.
• **Iterative**: Ascending clock auctions (except via custom dynamic loops). |
| **Matching Logic** | • **Bipartite**: One-to-one, one-to-many.
• **Stable Matching**: Deferred Acceptance logic.
• **Compatibility**: Explicit allow-/deny graphs. | • **General Graph**: Non-bipartite matching with long exchange cycles (e.g., kidney chains $> 2$). |
| **Dynamics & Stochasticity** | • **Distributions**: Poisson, Uniform, Custom discrete.
• **Randomness**: Drawing random values for valuations/supply.
• **Time Loops**: Multi-period execution with discounting.
• **Policy Functions**: Dynamic reserves/costs based on expressions. | • **Stochastic Optimization**: Solving *for* optimal policies under uncertainty (MIND simulates policies, it doesn't *solve* them).
• **Equilibrium Finding**: Computing Nash equilibria. |
| **Constraints** | • **Linear**: Budget, supply, capacity, fairness quotas.
• **Logical**: Mutual exclusivity, package constraints.
• **Incentive**: IR, IC (as linear constraints). | • **Non-Convex**: Integer constraints not mappable to MIP.
• **Black-box**: Constraints defined by external code. |
| **Objectives** | • **Convex**: Maximize Revenue, Welfare, Matches, or custom convex functions. | • **Non-Convex**: Deep neural network objectives.
• **Game-Theoretic**: "Maximize stability" (unless expressed as a matching rule). |

## B  FORMAL GRAMMAR (EBNF)

```
program             : "market" "{" market_block* "}" ;

market_block        : global_settings_block
                    | auction_block
                    | stage_block
                    | matching_block
                    | constraints_block
                    | objectives_block ;

global_settings_block
                    : "global_settings" "{" gs_element* "}" ;
gs_element          : currency_decl
                    | supply_decl
                    | reserve_price_decl ;

auction_block       : "auction" "{" auction_element* "}" ;
auction_element     : auction_type_decl
                    | participants_decl
                    | goods_decl
                    | valuations_decl
                    | allocation_rule_decl
                    | payment_rule_decl ;
```

```
stage_block        : "stage" "(" "name" "=" string_literal ")"
                     "{" stage_element* "}" ;
stage_element      : auction_block
                   | reaction_decl ;
reaction_decl      : "reaction" "(" "unsold_goods" "=" string_literal ","
                                   "auction_type"  "=" string_literal ")" ;

matching_block     : "matching" "{" matching_element* "}" ;
matching_element   : matching_type_decl
                   | participants_decl
                   | compatibility_graph_decl
                   | matching_rule_decl ;

constraints_block  : "constraints" "{" constraint_entry_list "}" ;
objectives_block   : "objectives"  "{" objective_entry_list  "}" ;

participants_decl  : "participants" "(" string_literal_list ")" ;
goods_decl         : "goods"        "(" string_literal_list ")" ;
valuations_decl    : "valuations"   "(" valuation_entry_list ")" ;

string_literal_list: "[" (string_literal ("," string_literal)*)? "]" ;
valuation_entry_list
                   : "{" (valuation_entry ("," valuation_entry)*)? "}" ;
valuation_entry    : string_literal ":" "{" good_value ("," good_value)* "}" ;
good_value         : string_literal ":" number ;

auction_type_decl  : "type"          "(" string_literal ")" ;
allocation_rule_decl
                   : "allocation_rule" "(" string_literal ")" ;
payment_rule_decl  : "payment_rule"  "(" string_literal ")" ;

matching_type_decl : "type"          "(" string_literal ")" ;
matching_rule_decl : "matching_rule" "(" string_literal ")" ;
compatibility_graph_decl
                   : "compatibility_graph" "(" compatibility_entry_list ")" ;
compatibility_entry_list
                   : "{" (compatibility_entry ("," compatibility_entry)*)? "}" ;
compatibility_entry: string_literal ":" "[" (string_literal ("," string_literal)*)? "]" ;

constraint_entry_list
                   : (constraint_param_entry | string_literal)
                     ("," (constraint_param_entry | string_literal))* ;
constraint_param_entry
                   : identifier "(" (parameter_assignment
                                    ("," parameter_assignment)*)? ")" ;
parameter_assignment
                   : identifier "=" value ;

objective_entry_list
                   : (string_literal ("," string_literal)*)? ;

string_literal     : ESCAPED_STRING ;
identifier         : /[A-Za-z_][A-Za-z0-9_]*/ ;
number             : SIGNED_NUMBER ;
value              : number | string_literal | boolean ;
boolean            : "true" | "false" ;
```

## C  VALIDATION

**What is verified**

- Names and References: unique goods/participants; auctions reference declared participants/-goods.

- Valuations Consistency: keys match auction participants; goods in valuations are declared; sparse entries → warnings.

- Rules Recognition: auction types, allocation/payment rules recognized or mapped from common aliases.

- Stage/Matching (if present): global settings sanity; stage naming; reaction fields; matching type/rule; graph nodes exist; symmetry warnings.

- Constraints/Objectives (if present): types recognized; basic parameter sanity; objective conflict warnings.

**Simple Validation Algorithm**

```
Input            : MarketProgram
Output           : ValidationReport (errors, warnings, suggestions); IR may be autofixed

1) Basic field checks (hard errors)
   - Good.name not empty; reserve_price >= 0
   - Bidder.name not empty; budget >= 0
   - Auction.auction_type not empty
   - Assignment fields not empty; bid_price >= 0

2) Core checks (always)
   - Unique names; auctions reference existing participants/goods
   - valuations match auction participants/goods; sparse -> WARN, mismatches -> ERROR
   - auction_type, allocation_rule, payment_rule:
     - map known aliases
     - unknown -> ERROR; some types partially implemented -> WARN

3) Stage/Matching checks (only if present)
   - Global settings: supply/reserve defaults; negatives -> ERROR, missing -> WARN
   - Stages: unique, named, each has an auction
   - Reauction: needs unsold_goods and auction_type; validate type; loose goods check
   - Matching: normalize matching_type; unknown -> WARN + default to bipartite
     - participants non-empty, no duplicates
     - compatibility_graph nodes exist; symmetry missing -> WARN
     - matching_rule: missing -> default stable_matching (WARN); unknown -> ERROR

4) Advanced checks (only if constraints/objectives present)
   - Constraints: type recognized; params sane (e.g., budgets >= 0, caps >= 0)
   - Objectives: normalize; unknown -> ERROR; conflicting goals -> WARN

5) Autofix safe defaults
   - Missing/unknown allocation_rule -> highest_bid
   - Missing/unknown payment_rule   -> second_price
   - Global supply missing/invalid  -> 1
   - Missing reserve_price (global/good) -> 0.0
   - All fixes logged as suggestions

6) Return report; program contains applied defaults where safe
```

## C.1 GOVERNANCE ARTIFACTS

To support auditing and compliance, MIND treats the compilation process as a governance event. A complete **Governance Artifact** is a versioned bundle containing four components that allow third parties to verify outcomes:

1. **Readable Specification:** The human-legible MIND DSL source code defining the mechanism rules.

2. **Typed IR Snapshot:** The serialized Intermediate Representation (AST) used by the compiler, serving as the single source of truth for semantics.

3. **Validator Report:** A structured log of all checks passed (parsing, typing, economic consistency) and any auto-fixes applied.

4. **Execution Integrity:** Cryptographic hashes of the spec and IR, logged alongside backend simulation outputs (allocations, payments) to ensure reproducibility.

Together, these components decouple the *intent* (DSL) from the *execution* (Backend), providing a transparent chain of custody for market mechanisms.

## D BACKEND SELECTION

**Heuristic (implemented in `MarketCompiler`)**

- Pure matching or simple Phase-1 auctions → Pandas (NetworkX for matching).

- Multi-stage without optimization features → Pandas + Prefect orchestration.

- Combinatorial auctions or constraints/objectives → CVXPY optimization.

**Mapping**

| IR features | Selected backend |
|---|---|
| auction only, valuations, simple rules | Pandas |
| matching (bipartite/stable) | Pandas + NetworkX |
| stage flow (no constraints/objectives) | Pandas + Prefect |
| constraints/objectives present | CVXPY |
| combinatorial auction | CVXPY |

# E  DATASET GENERATION PIPELINE

**Overview**

1. Sample a use case (914 total) and random market settings; assemble 4-shot prompt + grammar (markdown + EBNF).

2. Generate (GPT-4o-mini) brief description + DSL.

3. Filter with 4-stage validation (Parser → Validator → Compiler → Execute); keep only programs that pass all stages.

4. Enhance description (GPT-4o-mini) by extracting all facts from DSL; replace description text only.

5. Consistency check (GPT-4o): "YES/NO" whether description matches DSL; keep YES, drop NO.

6. Manual Inspection & De-duplication: Final manual review and strict de-duplication to form the final dataset.

## E.1  DATA FILTERING STATISTICS

To guarantee dataset quality, we tracked the number of samples retained at each stage. Table 5 details the rigorous filtering process.

Table 5: Dataset Filtering Pipeline Statistics. The high drop rate at the Consistency Check stage ensures that only accurately described mechanisms are retained.

| Pipeline Stage | Input Count | Dropped | Remaining |
|---|---|---|---|
| Raw Generation (GPT-4o-mini) | 20,108 | – | 20,108 |
| Stage 1 & 2: Parse + Validator | 20,108 | 551 | 19,557 |
| Stage 3: Compiler | 19,557 | 302 | 19,255 |
| Stage 4: Execute | 19,255 | 186 | 19,069 |
| Stage 5: LLM Consistency Check | 19,069 | 6,340 | 12,729 |
| **Validated Pool** | | | **12,729** |

---

**Prompt for data generation**

```
You are an expert Market Mechanism DSL generator. I will provide you with:
- The grammar of the MarketMechanismDSL,
- A few example DSL programs,
- A target use case,
- And a set of market settings.

Your task is to generate a complete MarketMechanismDSL program that
fits the given scenario and settings. Strictly follow the provided
grammar and take inspiration from the examples. Use the canonical
constructs: participants([...]), goods([...]), valuations({ ... }),
and valid allocation_rule/payment_rule names. Do not invent syntax
not present in the grammar.

Please output your response in EXACTLY the following format and nothing else:
```

```
Description:

```markdown
1-3 sentences written from the user's perspective describing the market
```

DSL code:

```dsl
<your complete MarketMechanismDSL program here>
```

---

Inputs:

DSL Grammar:
```markdown
{grammar}
```

DSL Program Examples:
```dsl
{examples}
```

Use Case and Settings:
- Domain: {domain}
- Scenario: {scenario}
- Settings: {settings}
```

Prompt for description completion (from DSL)

```
You are a precise technical writer. Given a MarketMechanismDSL program,
write a COMPLETE, human-readable task description that includes ALL facts
present in the DSL (participants, goods, valuations, auction/matching type,
allocation/payment/matching rules, key settings). Do NOT hallucinate new
entities or numbers. Use clear, concise prose (4-8 sentences).

Output EXACTLY in this format:

Description:

```markdown
<concise but complete description, entirely derived from the DSL>
```

---

Inputs:

DSL Program:
```dsl
{dsl_code}
```

Optional Examples (style guide only):
```markdown
{examples}
```
```

## F COMPLETENESS AGENT WORKFLOW

**Node-by-node extraction** Given a natural-language description, the agent queries nodes in order and marks each as *Enough/Not Enough*:

1. Auction (type, participants, goods, valuations, rules).

2. Global settings (currency, supply, reserve_price).

3. Stages (name, auction, optional reauction).

4. Matching (type, participants, compatibility_graph, rule).

5. Constraints (simple/parameterized).

6. Objectives (list).

If Not Enough, it asks for the missing facts before proceeding. Finally it merges fragments into a normalized schema and renders minimal DSL.

## G END-TO-END EXAMPLE (SIMPLE AUCTION)

### G.1 NATURAL LANGUAGE INPUT

**Step 1: Natural Language**

```
"Create a second-price auction for three participants competing for an advertising slot.
Alice values the slot at $100, Bob at $80,
and Carol at $90. Use highest bidder wins allocation."
```

### G.2 COPILOT GENERATION (NL → DSL)

**Step 2: Generated DSL**

```
market {
  auction {
    type("second_price")
    participants(["Alice", "Bob", "Carol"])
    goods(["AdSlot"])
    valuations({
      "Alice": {"AdSlot": 100},
      "Bob": {"AdSlot": 80},
      "Carol": {"AdSlot": 90}
    })
    allocation_rule("highest_bid")
    payment_rule("second_price")
  }
}
```

### G.3 PARSER (DSL → PARSE TREE)

**Step 3: Lark Parse Tree (excerpt)**

```
program
  statement
    auction_block
      auction_element
        auction_type_decl
          string_literal: "second_price"
      auction_element
        participants_decl
          string_literal_list
            string_literal: "Alice"
            string_literal: "Bob"
            string_literal: "Carol"
      ...
```

### G.4 TRANSFORMER (PARSE TREE → IR)

**Step 4: Intermediate Representation (MarketProgram)**

```
MarketProgram(
  goods=[Good(name="AdSlot")],
  bidders=[Bidder(name="Alice"), Bidder(name="Bob"), Bidder(name="Carol")],
  auctions=[Auction(
    auction_type="second_price",
```

```
     participants=["Alice","Bob","Carol"],
     goods=["AdSlot"],
     valuations={"Alice":{"AdSlot":100},"Bob":{"AdSlot":80},"Carol":{"AdSlot":90}},
     allocation_rule="highest_bid",
     payment_rule="second_price"
   )],
   assignments=[], global_settings=None, stages=[],
   matching=None, constraints=[], objectives=[]
)
```

## G.5  VALIDATION (IR SEMANTIC CHECK)

### Step 5: Validation

```
program.validate_comprehensive() successful
– All participants referenced in valuations
– All goods have valid valuations
– Auction type and payment rule compatible
– Unique participant and good names verified
```

## G.6  COMPILATION AND BACKEND SELECTION

### Step 6: Compiler Analysis

```
Analyzing DSL features...
– Single auction block detected
– No constraints or objectives
– No multi-stage features
– No matching markets

Backend Selected: Pandas
Reasoning: Simple auction, simulation-based approach sufficient
```

## G.7  EXECUTION (GENERATED CODE EXCERPT)

### Step 7: Pandas Backend Code (excerpt)

```python
# Allocation: highest_bid rule
allocations = {}
for good in goods:
    good_bids = bids_df[bids_df['good'] == good].copy()
    if not good_bids.empty:
        winner = good_bids.loc[good_bids['valuation'].idxmax()]
        allocations[good] = winner['participant']

# Payment: second_price rule
payments = {}
for good, winner in allocations.items():
    good_bids = bids_df[bids_df['good'] == good].copy()
    sorted_bids = good_bids.sort_values('valuation', ascending=False)
    if len(sorted_bids) >= 2:
        second_highest = sorted_bids.iloc[1]['valuation']
        payments[winner] = payments.get(winner, 0) + second_highest
    else:
        payments[winner] = payments.get(winner, 0) + sorted_bids.iloc[0]['valuation']
```

## G.8  RESULTS

### Step 8: Console Output

```
=== AUCTION RESULTS ===
Allocations:
  AdSlot: Alice
Payments:
```

```
   Alice: $90
Artifacts:
  - auction_results.csv
```

# H  ACADEMIC MECHANISM CASE STUDIES

Table 6 illustrates how MIND captures diverse mechanisms from recent literature. We map the core theoretical components of each paper to specific MIND language constructs.

Table 6: Mapping theoretical components to MIND constructs for 8 real-world mechanisms.

| Paper & Mechanism | Mapping to MIND Constructs | DSL Implementation (Snippet) |
|---|---|---|
| **Banchio & Skrzypacz (2022)** *Repeated First-Price Auction* Analyzes equilibrium in repeated auctions. | • **Repeated Game** → `stage` block with `discount`. • **Equilibrium Condition** → `threshold` w/ math expression. | `stage(name="RepeatedFPA") {` `auction {` `type("first_price")` `threshold(expr="(m-2)/(2m-3)")` `}` `discount(0.6)` `}` |
| **Gui et al. (2022)** *Dynamic Second-Price Auction* Updates participation probability dynamically. | • **Time Dynamics** → `dynamic` block with `periods`. • **Throttling Policy** → `cost_function` with `step()`. | `dynamic { periods(288) }` `auction {` `type("second_price")` `cost_function(` `target="participation_prob",` `expr="step(r_t,[0.3,0.5],[0.3,0.5])"` `)` `}` |
| **Feldman et al. (2022)** *Uniform Price Auction* Verifies welfare bounds. | • **Welfare Goal** → `objectives {"max_social_welfare"}`. • **Mechanism** → Standard `uniform_price` type. | `auction {` `type("uniform_price")` `allocation_rule("uniform")` `}` `objectives {` `"maximize_social_welfare"` `}` |
| **Goke et al. (2022)** *First-Price with Dynamic Floors* Real-time floor adjustment. | • **Price Floor** → `threshold` (reserve price). • **Adjustment Logic** → `cost_function` updating floor. | `auction {` `type("first_price")` `threshold(expr="dynamic_floor")` `cost_function(` `target="floor_policy",` `expr="update_rule"` `)` `}` |
| **Cashore et al. (2022)** *Stochastic Ridesharing (SSP)* Joint pricing and matching. | • **Rider-Driver Matching** → `matching` block (bipartite). • **Pricing Stage** → `stage` block with `auction`. | `matching {` `type("bipartite")` `matching_rule("maximize_matches")` `}` `stage(name="Pricing") {` `auction { type("second_price") }` `}` |
| **Anunrojwong et al. (2022)** *Robust Second-Price Auction* Regret minimization with reserves. | • **Regret Minimization** → Implicitly via `threshold` tuning. • **Reserve Price** → Parameterized `threshold`. | `auction {` `type("second_price")` `threshold(expr="reserve_price")` `}` `constraints {` `"incentive_compatibility"` `}` |
| **Ko & Munagala (2022)** *Optimal Randomized Auction* Revenue maximization under budget. | • **Budget Limit** → `constraints {"budget_constraint"}`. • **Optimal Goal** → `objectives {"maximize_revenue"}`. | `constraints {` `"budget_constraint"` `}` `objectives { "maximize_revenue" }` `stage(name="Stage1") {` `discount(0.6)` `}` |
| **Aouad & Ma (2023)** *Online Stochastic Matching* Matching with correlated arrivals. | • **Stochastic Demand** → `distribution` (Custom/Poisson). • **Expected Reward** → `cost_function` expression. | `matching {` `type("bipartite")` `distribution(name="demand",` `kind="Custom", params={...})` `cost_function(target="reward",` `expr="1-(1-rho)^2")` `}` |

# I  USER STUDY DETAILS

## I.1  METHODOLOGY

**Participants.** We recruited $N = 17$ participants with varying levels of programming and economic expertise. The breakdown of participants is as follows:

- **Education:** 13 undergraduate students, 4 graduate students (Master's/PhD).
- **Python Experience:** The cohort was technically proficient: 12 participants had $> 2$ years of experience, 2 had 1–2 years, and 3 had 0.5–1 year.
- **Domain Knowledge:** 10 participants reported prior familiarity with auctions or matching markets, while 7 reported no prior domain knowledge.

**Design.** The study followed a **within-subjects design**. Each participant performed modification tasks on the same three mechanisms in both **MIND** and **Python** environments. External tools (e.g., IDEs, documentation, AI assistants) were allowed to mimic realistic workflows, provided that the total time spent was reported.

**Tasks.** Participants started with working baseline code and were asked to implement specific policy modifications:

- **Task A (Pricing Rule):** Change the payment rule from second-price to first-price (pay-as-bid) while keeping allocation logic unchanged.
- **Task B (Multi-Stage & Reserve):** Introduce a global reserve price ($100) and a second "clearance" stage (reauction) for unsold goods.
- **Task C (Compatibility):** Adjust the bipartite matching constraints (e.g., restrict specific Buyer-Tutor pairs and add a new agent) without changing the matching algorithm.

## I.2  RESULTS ANALYSIS

Table 7 summarizes the performance metrics. MIND demonstrated a clear advantage in modification efficiency across all tasks.

Table 7: User Study Results ($N = 17$). Comparison of completion time (median minutes) and correctness rate between MIND and Python workflows.

| Task | Median Time (min) | | Speed-up | Correctness Rate (%) | |
|---|---|---|---|---|---|
| | **Python** | **MIND** | | **Python** | **MIND** |
| **Task A (Pricing)** | 10.0 | 1.0 | 10.0× | 88% | 100% |
| **Task B (Stages)** | 14.0 | 2.0 | 7.0× | 76% | 88% |
| **Task C (Matching)** | 8.0 | 2.0 | 4.0× | 82% | 88% |
| **Overall Average** | **10.7** | **1.7** | **6.3×** | **82%** | **92%** |

**Analysis.** The disparity was most pronounced in Task A and B. In Task A, MIND required only a keyword change, yielding a 10× speedup. In Task B, Python implementations often required significant refactoring of state management logic (14 min median), leading to higher time costs. In contrast, MIND allowed adding a stage in just 2 minutes. In Task C, direct manipulation of the `compatibility_graph` in MIND eliminated common index-error bugs observed in the Python group, improving both speed and correctness.

