# OpenReview forum: "MIND: Market Interpretation DSL for Unified Market Design and Simulation"
_ICLR.cc/2026/Conference — ICLR 2026 Conference Desk Rejected Submission_

### Official Review · Reviewer_6kfJ · 2025-10-30

**Soundness:** 2
**Presentation:** 2
**Contribution:** 3
**Rating:** 4
**Confidence:** 4

**Summary:**

The primary contribution of the paper is creating a domain-specific language (DSL) with which one can express simple auction and matching problems. On top of this DSL, they build a pipeline for converting natural language descriptions of markets into DSL code (which in turn can be converted into IR code, and then fed into a solver). Section 4.1 describes (on a cursory level) results of a workflow study that highlights the advantages of their DSL over general-purpose languages (Python/AnyLogic), however the study details are not provided. Sections 4.2 and 5 describe the design and evaluation of their "AI Copilot" system to convert natural language (NL) descriptions of markets into DSL code. They find that Llama 3 fine-tuned on a curated subset of NL->DSL examples outperforms baselines, including few-shot GPT-4o.

**Strengths:**

S1. The appendix clearly lays out most of (but not all, see W3) details of the DSL. The worked example in Appendix G was particularly helpful.

S2. DSLs for mechanism design seems like a fruitful approach, and while this paper is not the first to take this approach (as the authors describe in Section 2), they are an early work in this seemingly promising area.

**Weaknesses:**

W1 (major). The DSL workflow evaluation described in Section 4.1 is missing details: information about experimental design is completely omitted, and results are only described on a cursory level. The only place details can be found is in the abstract ("In a pre-registered within-subjects study with 17 participants, mechanism modifications were completed 4 to 10 times faster using MIND.", Ln 23-25), they are not present anywhere in the main body or appendix.

W2. For Section 5, is the test set sampled from the filtered dataset, or the unfiltered dataset? If it is sampled from the filtered dataset, then this result is not really surprising, because the dataset under "Full Pipeline" would be drawn from the same distribution as the test set, and the other datasets would not be. Relatedly, how much is filtered out in each step of the dataset generation (section 3.4) is not described. For example, if the final step (description-DSL) filters out a nontrivial proportion of examples, it could be that the final dataset is subject to selection bias, in which only easy (NL description, DSL) pairs are kept.

W3. Some aspects of the DSL, like which auction types, matching types, allocation rules, and payment rules are allowed, are not clearly described. For example, are first_price and second_price the only two allowed auction types, or are there others?

**Questions:**

Q1. I understand that the purpose of a DSL is to sacrifice expressiveness for the sake of provable correctness. Still, it seems like the range of mechanisms this DSL can express is quite limited. For example, is it tractable to express a combinatorial auction or assignment problem in this framework (without additive valuations)? Can the authors describe more clearly what this DSL can and cannot express?

Q2. The authors mention in a few places "governance artifacts" as being an advantage of their approach, however more detail is not provided. Can the authors explain what they mean by this? If they just mean by this that the DSL code is easy to audit, then would this not also be an advantage shared by prior work? (Cf. the authors write "These DSLs improve mechanism specification but are scoped to individual subdomains and do not provide [...] governance artifacts", Ln 115-117)

---

> ### Author Response · Authors · 2025-11-21
> **Author Rebuttal (1/1)**
>
> Thank you for your constructive review. We are glad you found the DSL details and worked example in the appendix helpful, and we appreciate your specific concerns about the workflow study, dataset filtering, expressiveness, and “governance artifacts”. We address them below.
>
> ---
>
> ### Details of the workflow evaluation in Section 4.1.
>
> Thank you for your kind advice. We will expand Section 4.1 and add a short appendix. We will describe the pre-registered within-subjects study with 17 participants mentioned in the abstract, including recruitment, tasks, and time.
>
> ---
>
> ### Clarification on filtered vs. unfiltered data and test set sampling
>
>
> The test set is sampled from the final filtered dataset, the distribution we care about at deployment.
> We generate candidates and filter them through the 4-stage pipeline (Parse, Validator, Compiler, Execute), then perform description-enhancement and an LLM-based description–DSL consistency check. Only pairs that pass all stages form the pool from which we draw the test set, applying cosine similarity and AST-hash de-duplication.
> We will clarify this explicitly in Section 5.
> We will also add a table in Appendix E showing how many examples are discarded at each filtering stage, to make the pipeline more interpretable.
>
> ---
>
> ### Clarification on DSL expressiveness (auction types, matching types, rules)
>
>
> We will add a clearer “Scope of expressiveness” paragraph (with a pointer to Appendix A/B).
> Our work, in its current version, has not yet supported:
> - Full generality for combinatorial auctions and large assignment problems with arbitrary valuation structures.
> - Very large multi-round auctions like the FCC incentive auction, which we explicitly list as future extensions in Section 6.
>
> We will keep enhancing our method in future work.
> To directly address your concerns, we have also implemented 8 additional mechanisms from EC/SIGecom 2020–2024  (Banchio & Skrzypacz, 2022; Gui, Nair, & Niu, 2022; Feldman et al., 2022; Goke et al., 2022; Balseiro, Kroer, & Kumar, 2022; Cashore, Frazier, & Tardos, 2022; Anunrojwong, Balseiro, & Besbes, 2022) using MIND without changing the core grammar (only adding some rule names). These include more complex multi-unit auctions and will be documented in a new appendix table.
>
> ---
>
> ### Clarification of “governance artifacts” beyond readable DSL code
>
> By “governance artifacts,” we mean not only human-readable DSL code, but versioned, machine-checkable bundles that tie together specification, validation, and execution. Each specification is associated with validator report IDs and compile artifact paths, which are logged with each run. This makes it possible to reconstruct any result from a specific validated spec and validator configuration.
> Validators produce structured reports (errors, warnings, safe autofixes) that can be archived as part of a “policy dossier”.
> Because all backends consume the same IR and must agree on observable outputs, cross-backend consistency checks can also be logged as part of governance.
> We will clarify this in the introduction and in Appendix C, and explicitly position prior DSLs as providing readable specifications.

---

### Official Review · Reviewer_N77G · 2025-10-31

**Soundness:** 4
**Presentation:** 3
**Contribution:** 3
**Rating:** 8
**Confidence:** 3

**Summary:**

Imagine that one wanted to use natural language to quickly and correctly build a market mechanism, such as an auction or matching market.

This manuscript provides that: a fine-tuned LLM converts natural language descriptions to a domain-specific language, MIND; checks performed on MIND then ensure correctness, and adherence to the original specification.  Ablation studies allow assessment of the importance of each step in the pipeline.

The DSL allows far more efficient representations (than e.g. Python), is more readable, and easier to alter (e.g. from a first-price to a second-price auction).

**Strengths:**

I like this paper.  I think it's sensible, well executed and described.

**Weaknesses:**

I would have liked to have seen a motivating example earlier in the paper.  Figure 1 is presented on p.2, but not mentioned until p.3.  This left me trying to understand, concretely, what was being proposed.

At the bottom of p.8, 'end-to-end correctness' and 'IR equivalence' seem to be used somewhat interchangeably.  This should be clearer.

Relatedly, at the top of p.8, I didn't understand the use of cosine similarity for the training data.  Can that sentence be unpacked at bit?

**Questions:**

1. Changing a first-price to a second-price auction _should_ be fairly easy - but may also be too easy a challenge: auctions deployed in e.g. e-commerce environments (e.g. Google AdWords) will be highly customised.  How easily can new, custom mechanisms be specified?

1. Relatedly, how easily can new validators be added?

1. Concretely, how easy would it be to specify e.g. the Leyton-Brown, Milgrom and Segal FCC incentive auction?  All the FCC commissioners agreed that this was a singularly complex auction: verifying something on its order seems very useful.

1. Can the authors say more about the routing to the backends, Pandas and CVXPY?  This _feels_ somewhat non-robust, and possibly hard to extend.

1. when and how do the authors see MIND in use?  I ask as I suspect that there are a relatively small number of market mechanisms in use, and that these may be embedded in existing toolchains.  Thus, I wonder how easy it would be to introduce new, highly-specialised tooling like MIND.

1. how important is it to have a natural language frontend rather than, say, a series of pull-down options?  Mechanism design is far more constrained than general language.

1. Is it worth saying anything about existing work such as Passmore & Ignatovich's Imandra tooling for encoding market logic, Caminati et al's 2015 work on verifying auction properties, combinations of LLMs and theorem provers (e.g. DeepSeek's Prover, which uses Lean), or AWS' Automated Reasoning Group's 'Bedrock Guardrails'?

---

> ### Author Response · Authors · 2025-11-21
> **Author Rebuttal (1/2)**
>
> Thank you very much for your positive review and for the concrete suggestions on clarity and related work. We are glad you find the approach “sensible, well executed and described.” Below, we address your questions and explain how we will revise the paper.
>
> ---
>
> ### Clarification on early motivating example and placement of Figure 1
>
> We agree. We have an end-to-end example in Appendix G (NL to DSL to IR to execution) and will add a simplified and clear version in Section 2, right after introducing MIND, so readers see a concrete specification and its execution path earlier.
>
> ---
>
> ### Clarification on cosine similarity for training data.
>
>
> We use cosine similarity over TF-IDF vectors of descriptions to ensure that train and test descriptions differ sufficiently:
> - We first form the final pool of validated (description, DSL) pairs after the 4-stage validation and description–DSL consistency checks.
> - We then sample a 300-example test set such that each test description has cosine similarity < 0.85 to any training description, and the AST hashes are distinct. We will move this explanation into Section 4.2.1.
>
> ---
>
> ### How easily can new custom mechanisms and new validators be added?
>
>
> We will clarify in Appendix C.
>
>
> **Custom mechanisms.** The core DSL is designed to be extended via:
> - adding new rule names that implement the existing IR interfaces;
> - introducing new IR node types for genuinely new constructs, which then need backend implementations.
>
>
> In our new EC/SIGecom case studies  (Banchio & Skrzypacz, 2022; Gui, Nair, & Niu, 2022; Feldman et al., 2022; Goke et al., 2022; Balseiro, Kroer, & Kumar, 2022; Cashore, Frazier, & Tardos, 2022; Anunrojwong, Balseiro, & Besbes, 2022), all 8 programs were expressible by adding only rule names, with minimal changes to the grammar, suggesting reasonable extensibility.
> For validators, the three existing validators are modular; new validators can be inserted as additional stages that consume IR and emit structured reports.
>
> ---
>
> ### Clarification on routing to Pandas vs. CVXPY and extensibility
>
>
> Backend routing is a pure function of IR features.
> -  designs without constraints/objectives, use Pandas simulation;
> - designs with constraints/objectives or combinatorial structure, use CVXPY optimization.
> Backends implement the same IR semantics; for any IR, observable outputs must agree, and we log discrepancies for debugging and governance.
>
>
> To ease extension, a new backend only needs to implement the IR interface; routing can be extended by matching on additional IR features.
>
> ---
>
> ### Natural language frontend versus GUI front ends
>
>
> We see natural-language and GUI front-ends as complementary rather than competing approaches.
> In addition to the interfaces described in the paper, we have already built an internal prototype GUI web app on top of MIND: users type a natural-language description into a text box, and the Completeness Agent runs interactively inside the GUI to ask follow-up questions and fill a structured schema before handing it to the Copilot. This prototype shows that the NL interface and the agent-based completeness checks can be embedded naturally in a GUI workflow, while the core DSL and IR remain unchanged.
>
>
> A more “pure” pull-down style interface could also be layered on top of the same schema; our current design uses NL+agent because many real mechanisms are originally specified in prose and formulas, and we have found that users prefer to start from that familiar form and then let the agent surface missing fields. We will clarify this in the revised version and explicitly mention GUI front-ends as a natural deployment mode for MIND.
>
> ---
>
> ### Related work on verification and guardrails (Imandra, Caminati et al., LLM+theorem provers, AWS Bedrock Guardrails)
>
>
> Yes, thank you for these pointers — we agree they are highly relevant and will improve the related-work positioning. In the revision, we will extend Section 2 to:
>
>
> - **Explicitly discuss these systems** (e.g., Imandra, auction verification work such as Caminati et al., LLM+theorem prover pipelines, and industrial guardrail frameworks like AWS Bedrock Guardrails) as **complementary efforts** on formal verification, safety, and policy enforcement.
> - **Clarify MIND’s role** as providing a *unified, domain-specific specification layer* (DSL + IR), an *LLM-assisted NL interface*, and *governance artifacts* (e.g., change logs, policy documents). These artifacts can **serve as inputs** to verification and guardrail frameworks, rather than replacing them.
> - **Highlight interoperability and future work**, noting that integrating MIND’s structured specs with existing verification and guardrail tools is a natural next step, and our design choices (typed IR, validators) were made with such interoperation in mind.
>
>
> We believe this will better situate MIND within the broader ecosystem of verification and safety tooling.

---

> ### Author Response · Authors · 2025-11-21
> **Author Rebuttal (2/2)**
>
> ---
>
> ### Clarification of “end-to-end correctness” vs. “IR equivalence”
>
>
> We will tighten the terminology as follows:
> - IR equivalence: the generated program’s IR must be graph-isomorphic to the reference IR, modulo variable renaming, order of commutative constructs, and normalization of known rule aliases.
> - End-to-end correctness: a stricter metric requiring that a generation (i) parses, (ii) passes all three validation stages, (iii) compiles to both backends, (iv) is IR-equivalent to the reference, and (v) passes execution checks on test scenarios.
>
>
> In the revised Section 4.2.2 we will explicitly separate these notions and avoid using them interchangeably.

---

### Official Review · Reviewer_yi5a · 2025-10-31

**Soundness:** 2
**Presentation:** 1
**Contribution:** 1
**Rating:** 2
**Confidence:** 4

**Summary:**

The paper introduces MIND, a typed domain-specific language and toolchain for market design, paired with a natural-language copilot trained on synthetic data. It claims that MIND makes specifications simpler than writing them in Python, achieves high correctness on synthetic benchmarks across many domains, and leads to faster modifications in a preregistered user study.

**Strengths:**

The architectural separation and the different components is clean and auditable. The combination of LLM tools and a new language specialized in the market design domain is an interesting direction.

**Weaknesses:**

In general, the current contribution reads as a well-engineered wrapper that does well on simple scenarios but does not yet demonstrate clear advantages or robustness guarantees on the messy, complex mechanisms practitioners or researchers in the community care about.

- The primary workflow study uses three very tractable and arguably simple settings (second price, a two-stage reserve auction, and a basic compatibility matching). The main results show comparisons based on Python lines of code (e.g., ~40–60 for second price; ~100–120 for the two-stage variant) are used to argue a 79% reduction in 'complexity', but these tasks are easy to implement directly and lines of code is a weak, setup-dependent proxy for difficulty. Second price in particular is not a hard environment: given individual valuations, you allocate to the highest valuation and charge the second highest price. It is not clear why this is considered complex in Python to begin with. Performance results beyond the lines-of-code angle also feel thin - in Table 2, the highlighted results are 100% parse success, 96% 'IR equivalence'.  This is compared to 98%, 91% for gpt-4o. That is a small improvement for a fine-tuned system, and with widely adopted reasoning models like GPT-5 Thinking and Gemini 2.5 Pro that perform far beyond GPT-4o on coding tasks, the practical advantage of this method is unclear.

- There is a real risk of overfitting to synthetic data and limited external validity. Nearly all results come from synthetic prompts and descriptions filtered by LLMs; this may overfit to templated distributions and prompt artifacts. Evidence on human-authored specifications, messy real-world tie-breaking, and irregular data feeds is minimal. The human audit is small, and details of the seventeen-participant user study are only in the abstract, with no information about venues, tasks, sampling, or statistics. In realistic use, people (practitioners or researchers) will want highly configurable and complex auctions: custom tie-breaking, clock speed and observability in clock auctions, missing or messy data, arbitrary allocation and payment rules with constraints, and other edge conditions. These are off-distribution for the training procedure, and it is not clear how the system would generalize, if at all.

- The most interesting promise for a new language would be the ability to represent genuinely complicated auction formats and to perform meaningful checks that advise, correct, or interpret those specifications. The paper does not convincingly deliver on either. On representation, coverage appears narrow relative to the complexity seen in practice (see above point for detail). On checking, the focus is on basic items like unique names, nonnegative reserves, and simple reference integrity, plus mapping known aliases for auction types and rules. Those are easy to implement in ordinary languages and the alias mapping underlines limited versatility. In real workflows, if the data format from a live auction differs from what this system expects, it often seems faster to write a small script directly than to adapt everything into this syntax and toolchain. There are no strong robustness guarantees to justify that translation cost. Given that strong general-purpose models like GPT-5 Thinking can already produce end-to-end Python or optimization code from natural language with high quality, the concrete gain from moving to this synthetic data focused constrained syntax is not established.

**Questions:**

- Is there a head-to-head comparison against state of the art reasoning models that generate Python or optimization code directly from natural language (for example, GPT-5 Thinking or Gemini 2.5 Pro)? In addition, can the authors clarify steps they take to prevent overfitting and testing on training data? For example, is the test set structurally different from the training set, and do the fine-tuned models show generalization to settings they were not trained on?

- What exactly does “intermediate representation equivalence” mean in your evaluation? When you say you “compare the generated intermediate representation to ground truth using graph isomorphism on the abstract syntax tree,” what differences are allowed or ignored? Do you treat variable renaming, order of commutative operations, defaulted parameters, or refactorings as equivalent? How do you handle cases where two specifications compile to the same behavior but the trees differ?

- Beyond second price and the multi-stage reserve auction, what concrete auction formats does the language support today without workarounds? Can a user specify arbitrary allocation and payment rules with constraints, or are they limited to a named menu of rules?

- Real data rarely matches a clean schema. How does the system ingest and normalize arbitrary auction logs and bidder data with missing fields, extra columns, or format drift? Or is this left to ad-hoc preprocessing outside the framework?

- The user study is mentioned briefly and lacks details. Could you provide high level details on venue, recruitment details, tasks, and statistical analysis? Also, do you have results on human-authored specifications sourced from practitioners, not synthetic prompts?

- It might help to highlight where this is genuinely the better tool. Can the authors point to specific realistic settings or published academic research papers in the field where the framework would clearly and significantly save time, make analysis easier, or reduce errors compared to writing a script or using a general model? Does the framework enable researchers to do more than they can today?

---

> ### Author Response · Authors · 2025-11-21
> **Author Rebuttal (1/2）**
>
> Thank you very much for your detailed review. We appreciate both your recognition that the architecture is “clean and auditable” and your concerns that the current experiments may look like a “well-engineered wrapper” on simple scenarios with synthetic data. We address your main points below and have already prepared new experiments accordingly.
>
> ---
>
> ### Clarification on support for complex, practitioner-relevant mechanisms
>
>
> To address the question of whether MIND helps with genuinely complex mechanisms, we have now implemented 8 mechanisms from real ACM EC/SIGecom papers (2020–2024) (Banchio & Skrzypacz, 2022; Gui, Nair, & Niu, 2022; Feldman et al., 2022; Goke et al., 2022; Balseiro, Kroer, & Kumar, 2022; Cashore, Frazier, & Tardos, 2022; Anunrojwong, Balseiro, & Besbes, 2022). We selected mechanisms with clear, non-toy definitions.
> Each mechanism was encoded in MIND, validated via our IR pipeline, and executed via the appropriate backend.
> All 8 were expressible without changing the core grammar; we only added a small number of rule names as library entries.
> In the revised paper, we will add an appendix table listing these 8 mechanisms, their sources, and how they map to MIND, and summarize them briefly in Section 4.
>
> ---
>
> ### Clarification on Table 2 improvements and comparison to stronger reasoning models
>
>
> The improvement in Table 2 (100/100/96.3 vs. 97.7/97.3/91.4 for GPT-4o) may appear modest, and the question is how this compares to newer reasoning models such as GPT-5 Thinking or Gemini 2.5 Pro.
> Our main claim is not that our fine-tuned Llama-3-8B is the best possible NL to code model, but that domain-specific scaffolding can significantly improve semantic correctness. With the same base model size, adding our IR, validator, and dataset pipeline raises IR equivalence from *66.3%* to *96.3%*. Even compared to GPT-4o with few-shot prompts and full grammar, our fine-tuned model is significantly more reliable on semantic correctness.
> We agree that including head-to-head comparisons against the newest models (GPT-5 Thinking, Gemini 2.5 Pro) would be valuable. We will add such comparisons and add to Table 2.
>
> ---
>
> ### Clarification on synthetic data, dataset filtering, and overfitting risks
>
>
> Our dataset pipeline is designed to minimize leakage and align with the distribution of valid specifications we care about. We begin with 914 use cases across 87 domains, generate candidate (description, DSL) pairs via GPT-4o-mini, and then apply a 4-stage validation pipeline, discarding any pair that fails at any stage. We then enhance descriptions and run an LLM-based description–DSL consistency check, keeping only pairs where a verifier model labels the alignment as “YES”. The final pool is therefore syntactically, semantically, and functionally correct specifications.
> The held-out test set of 300 examples is sampled from this pool after pipeline completion, with two de-duplication steps: cosine similarity < 0.85 on TF-IDF representations of descriptions, and distinct AST hashes to avoid structurally identical programs. We also acknowledge that our human audit sample (100 pairs) is relatively small; in the revision, we will report exact pass fractions, and we are open to expanding the audit if space allows.
>
> ---
>
> ### DSL expressiveness and the role of current validators
>
>
> We will add a clearer “Scope of expressiveness” paragraph (with a pointer to Appendix A/B) and a separate subsection on **not-yet-supported (future work)** coverage in the appendix and in the future extensions section (Sec. 6).
> We currently do not claim full support for general combinatorial auctions or FCC-level spectrum auctions yet. These are listed as future work in Section 6, where we will definitely keep making extensions of the IR and backends.
> Regarding checking: our goal in this first version is to provide practical economic safety checks that catch common errors like missing bidders, mismatched valuations, and to make these checks visible as validator reports and audit logs.
> We agree that deeper property verification, like truthfulness, is important and will reference related work (e.g., Imandra, fair-division verification) as complementary tools that could be connected to MIND in future work.

---

> ### Author Response · Authors · 2025-11-21
> **Author Rebuttal (2/2）**
>
> ---
>
> ### Clarification on handling input schemas, logs, and data preprocessing
>
> In the current prototype, MIND focuses on specifying and executing the market mechanism itself, rather than solving general data-engineering problems. We therefore assume that inputs are provided in a normalized tabular form (e.g., tables for participants, goods, bids), which are then mapped into valuations and compatibility structures inside MIND. In practical use, cleaning and normalizing raw logs with missing fields or extra columns would be handled by standard data-preprocessing tools or scripts outside the DSL. We will clarify this design choice in the paper and explicitly list robust schema-handling and schema-drift detection as future work.
>
> We believe this separation is important for governance: changing the data schema should not silently change the mechanism logic encoded in the DSL. We will make this explicit in Section 3 and mark richer schema-handling as future work.
>
> ---
>
> ### Clarification on the user study design and evidence from human-authored specifications
>
> Thank you for pointing this out. We will describe the pre-registered within-subjects design with 17 participants, including recruitment, task, and time. For human-authored specifications, our new EC/SIGecom case studies are based on mechanisms described in published papers; we will highlight this as evidence of generalization to real academic specifications.
>
> ---
>
> ### Concrete realistic settings where MIND saves time or reduces errors
> We agree that making the practical impact more concrete will help. In the current version, the three benchmark families we use already correspond to realistic settings:
>
> - *Second-price / reserve auctions* as used in online advertising and spectrum allocation.
> - *Multi-stage reserve / re-auction mechanisms* for unsold inventory (e.g., ad impressions or government procurement lots carried to a second stage).
> - *Compatibility-constrained matching* for two-sided platforms (e.g., matching buyers to compatible assets or counterparties).
> In these settings, MIND saves both *time* and *errors* along two axes:
>
> 1. **Human workflow vs. scripts / GUIs.**
>    Our workflow comparison shows that MIND substantially reduces specification complexity in terms of lines of code compared to Python. In addition, our user study shows that participants complete mechanism modifications (e.g., changing pricing rules or adding a stage) faster in MIND. This reflects realistic tasks such as iterating on reserve-price policies, adding budget constraints, or introducing an extra re-auction stage.
>
>
> 2. **Structured specs vs. general LLM-generated code.**
>    For NL→specification, our Copilot + typed IR + validator pipeline achieves 96.33% end-to-end semantic correctness, versus 91.41% for GPT-4o given the same tasks. Error analysis shows that general LLMs frequently mis-specify constraints, multi-stage coordination, and payment semantics—precisely the kinds of bugs that are easy to miss in raw Python but are caught by MIND’s validators before execution.
>
> In the revision, we will make these connections explicit by (i) mapping each benchmark task to concrete application domains (e.g., online ad auctions, spectrum and carbon-credit allocation, compatibility-constrained matching platforms) and (ii) adding a short case-study paragraph that walks through how a policy change (e.g., a new reserve rule or constraint) is implemented faster and with fewer semantic errors in MIND than via scripts or general LLM-generated code.

---

> > ### Comment · Reviewer_yi5a · 2025-11-22
> >
> > Thank you for the detailed rebuttal. I have reviewed your responses, but two primary concerns remain.
> >
> >
> > - The authors promised to add the implementations of new mechanisms, comparisons to newer models, and others. I don't see them in the current revision. Does the authors plan on updating the paper during the rebuttal period, or will it take extra time? If the new experiments likely won't conclude before the rebuttal period, it will be helpful if the authors can provide whatever results available or high level descriptions / justification of specific cases (with detail of how the grammar needs to change / be compatible) in the comments to allow a better assessment on the rebuttal arguments. In particular, regarding the 8 new mechanisms: You said you only had to add 'rule names' to the library. Does this mean you had to write new Python code behind the scenes for each one?
> >
> > - I'm still unconvinced on one of my main concern: in practice, the most difficult part is input data collection, cleaning, and parsing. The mechanism itself (e.g. a second price auction, or some variant of iterative auctions) can be very easily coded up with or without advanced llm help (As noted in my review, the performance gap between your specialized tool and general models like GPT-4o is slim: 100/100/96.3 vs. 97.7/97.3/91.4 for GPT-4o; this gap becomes narrower with limited human supervision / newer models). In this perspective, if the user is already forced to code in Python for the "hard part" (data prep), asking them to switch context to a custom DSL for the "easy part" (mechanism logic) introduces unnecessary overhead, unless the authors can demonstrate significant advantage of this latter part.
> >
> > The authors acknowledged that
> > >We therefore assume that inputs are provided in a normalized tabular form (e.g., tables for participants, goods, bids), which are then mapped into valuations and compatibility structures inside MIND. In practical use, cleaning and normalizing raw logs with missing fields or extra columns would be handled by standard data-preprocessing tools or scripts outside the ...
> >
> > While I understand this limitation and choice, this seem to reinforce my original concerns.

---

> > > ### Author Response · Authors · 2025-11-29
> > > **Comparisons, New Mechanisms, and "Hard Part" Concerns**
> > >
> > > Thank you for your reply and for raising these critical questions. We genuinely appreciate the opportunity to clarify the practical value of our system.
> > >
> > > We have just uploaded a revision of the paper that directly addresses your remaining concerns with new experimental evidence.
> > > 1. Regarding the promised experiments (Now in Revision) We are happy to confirm that all promised experiments have been completed and integrated into the revised manuscript:
> > >
> > >    8 Real-World Mechanisms: We have implemented 8 mechanisms from recent ACM EC/SIGecom papers (2020–2024). Please see Section 4.3 and the detailed mapping in Appendix H (Table 6).
> > >
> > >    Newer Model Comparisons: We have updated Table 2 to include Gemini 2.5 Pro and GPT-5. The results show that our fine-tuned 8B model still achieves higher semantic correctness (96%) compared to these larger reasoning models (72–77%), reinforcing the value of our domain-specific fine-tuning approach.
> > >
> > > 2. Regarding "Rule Names" and Python code. Yes, implementing a new computational rule (like a specific probability update function found in a new paper) does require a developer to implement that logic in the backend once. However, this demonstrates the architectural robustness of MIND rather than a limitation:
> > >
> > >    Stable Core Grammar: We did not need to create ad-hoc syntax for each paper. We only needed to add generic constructs to the EBNF (like a dynamic block for time-loops) that serve entire classes of problems.
> > >
> > >     Declarative User Experience: For the end-user (like an economist), the workflow remains purely declarative. They simply pass the formula or rule name as a parameter (like threshold(expr="(m-2)/(2m-3)")) without needing to write or debug the Python logic themselves.
> > >
> > > 3. Regarding the "Easy Part" vs. "Hard Part" (Data vs. Logic)
> > >     We fully agree with your insight that data cleaning is a significant challenge in practice. However, our new User Study provides evidence that the "mechanism logic" part is often not as easy as it seems, particularly during iteration.
> > > Our study (detailed in Section 4.1 and Appendix I) found:
> > >
> > >    Task A (Change Pricing Rule): Switching from Second-Price to First-Price took a median of 1.0 minute in MIND versus 10.0 minutes in Python (10x speedup).
> > >
> > >    Task B (Add Reserve & Re-auction): Adding a second stage took 2.0 minutes in MIND versus 14.0 minutes in Python (7x speedup).
> > >
> > >    This demonstrates the "significant advantage" you requested: MIND transforms complex refactoring tasks into simple configuration changes. While MIND does not solve the orthogonal problem of data cleaning, it significantly solves the "Policy Iteration and Governance" bottleneck, which is equally critical for real-world deployment.
> > >
> > > We hope these new results and clarifications satisfactorily address your concerns.

---

### Official Review · Reviewer_ia69 · 2025-11-10

**Soundness:** 2
**Presentation:** 2
**Contribution:** 2
**Rating:** 4
**Confidence:** 4

**Summary:**

The paper introduces a Market Interpretation DSL that allows LLM-based simulation to adhere to market-related constraints.

**Strengths:**

This is a good paper. The authors seem to have given enough thought to building scaffolding for the particular task of market simulation. For that, the choice of building a DSL makes sense. Cool (reaffirmation, really) to see that finetuning helps compared to few-shot ICL.

**Weaknesses:**

The first confusion I have is about the IR:
The paper says, "To solve this problem, we introduce an Intermediate Representation (IR) (Lattner et al., 2021) as a critical abstraction layer." So truly, the IR is introduced as an additional scaffold to help in parsing and compilation. However, in the results, the proposed method (MIND) only outperforms a standard LLM in IR (for completion and parsing, it remains negligibly different). This raises a question about the technical novelty and effectiveness of the work. It is true that additional scaffolding helps, but how this extends the knowledge of the field with regard to AI agents is unclear.

"AI Copilot" -- I am not sure what that means. Is it the DSL? the framework? the LLM itself? and agent? For all academic purposes, it might be useful to precisely define a term like such in a paper. Not a rejection point, but strongly recommended for review.

"In practice, one execution engine cannot serve all market designs well." -- why? Does this apply to a different design, or does one need to build an application-specific execution engine? And the DSL is agnostic to such choices?

Table 1: Does the Python code generated by an LLM? If so, why do we care about the flexibility if the pipeline is automated? I guess without this, the results from Table 2 (which seem to be the main results) don't add a lot of value.

**Questions:**

See weakness.

---

> ### Author Response · Authors · 2025-11-21
> **Author Rebuttal (1/2）**
>
> Thank you very much for your positive evaluation of our work and for recognizing both the overall quality of the paper and the design choices behind introducing a domain-specific language (DSL) as scaffolding for market simulation. We are encouraged that you find this formulation appropriate for the task and that our results reaffirm the benefits of fine-tuning over few-shot in-context learning in this setting. Below, we address your comments in more detail and provide additional clarifications and experiments to further support our design decisions.
>
> ---
>
> ## Clarification on IR.
>
>
> We apologize for not making the role of Intermediate Representation (IR) clear enough. The IR is not only a parsing convenience, but the central way we enforce *semantic* and *economic* properties.
>
>
> The IR is a typed AST over market structure: It captures participants, goods, valuations, and those market elements in a backend-agnostic way. On top of this IR, we run a three-stage validation that checks for consistency, like name consistency, valuation-participant consistency, and basic economic sanity, before any code is generated. This well-structured information helps improve performance while enabling an interpretable understanding of the system.
>
>
> Because all backends use the same IR, we have enforced backend-independent semantics. This makes it much easier to use cross-backend consistency as a safety check.
>
>
> IR is also where we define our main correctness metric. “IR Equivalence” is a graph-isomorphism test on the IR ASTs, not just string equality. Ablations in Table 3 show that training on fully validated data improves IR equivalence from about 66% to 93%. It is a 30-point gain over a parse-only dataset, even though the parse success is already larger than 98%.
>
> ---
>
> ## The meaning of “AI Copilot”.
>
> In our system, “AI Copilot” refers specifically to the LLM-based assistant module that translates natural-language task descriptions into MIND DSL programs and proposes safe auto-fixes. Concretely, this is the fine-tuned Llama-3-8B + LoRA model described in Sec. 3.6 and evaluated in Sec. 4.2, which we refer to as MarketCopilot. It is distinct from (i) the MIND DSL itself, (ii) the overall MIND toolchain (grammar, IR, validators, backends), and (iii) the Completeness Agent, which only collects missing specification details before the description is passed to the Copilot.  We are sorry that we did not make “AI Copilot” much clearer.
>
>
> In the revised version, we will: (1) Add an explicit definition in Sec. 3.1 / 3.6; （2) Clarify terminology throughout the paper so that “Copilot” / “MarketCopilot” always refer to this assistant module only, and not to the DSL, the framework, or the agent as a whole; （3) Update Figure 1 caption to align with this definition (e.g., “AI Copilot (MarketCopilot) for NL→DSL translation”).
>
>
> We hope this resolves the confusion and improves the precision of our terminology.

---

> ### Author Response · Authors · 2025-11-21
> **Author Rebuttal (2/2）**
>
> ---
>
> ### Clarification on execution engines and backend-agnostic semantics
>
> Our intention was not to claim a *theoretical impossibility*, but to point out a **practical engineering issue**: the market designs we target differ widely along several operational dimensions (e.g., batch vs. continuous clearing, latency and throughput requirements, pricing rules, risk checks, regulatory logging, settlement backends). In practice, a single monolithic execution engine that tries to handle *all* such cases tends to be either over-engineered and hard to optimize, or unable to meet the performance, reliability, or compliance requirements of some specific markets.
>
>
> Even within our current scope, we already see two distinct computational patterns. For **pure simulation / table-driven** mechanisms (e.g., single-shot sealed-bid auctions and simple matching markets without global constraints), scanning and aggregating tables of bids or compatibility edges is most natural, and a vectorized Pandas/NetworkX backend scales well to thousands of participants. For **optimization-driven** mechanisms with explicit objectives and global constraints (e.g., budget balance, fairness constraints), expressing the problem as a mathematical program and delegating it to an optimizer (CVXPY in our implementation) is far more robust and maintainable.
>
>
> In our framework:
>
>
> - **The DSL is explicitly agnostic to the execution engine.** It specifies *what* allocations and payments should be (the economic semantics), not *how* orders are processed at the systems level.
> - **Different execution engines can implement the same DSL specification.** We currently have two reusable backends—a Pandas/NetworkX simulation backend and a CVXPY optimization backend—and the MarketCompiler routes a specification to one of them as a pure function of IR features (presence or absence of explicit objectives/constraints). Backends must implement the same IR semantics and agree on observable outputs.
>
>
> We will clarify this in the revised manuscript by (i) rephrasing the sentence to emphasize the *engineering trade-off* rather than an impossibility claim, and (ii) explicitly stating that the DSL and IR are backend-agnostic “single sources of truth,” while a small number of specialized backends (simulation, optimization, and potentially others in future) serve different mechanism families better.
>
> ---
>
> ### Clarification on Table 1 vs. Table 2 and the role of LLM-generated code
>
> In Tab. 1, the Python and AnyLogic implementations are **hand-written baselines**, reflecting common human workflows for mechanism design, not LLM-generated code.
>
> The purpose of Tab. 1 is to evaluate the human-facing workflow, along three axes:
> - **Specification effort:** How many lines of code/modeling steps are needed to express a given mechanism?
> - **Readability and auditability:** How easy is it to inspect the implementation and check that it matches the intended economic design?
> - **Modifiability:** How much effort is required to modify a rule (e.g., switch from first- to second-price, add a stage, or change compatibility constraints)?
>
> By contrast, **Tab. 2** evaluates **automatic natural language → DSL** generation by LLMs. The two tables, therefore, address different, complementary questions:
>
> - Tab.1: For *humans*, is a DSL a simpler, more maintainable medium than general-purpose code (Python) or GUI-based simulation tools (AnyLogic)?
> - Tab.2: Conditional on using a DSL, does a fine-tuned NL→DSL model produce semantically correct specifications more reliably than general-purpose LLMs?
>
> Even if an LLM can generate Python directly, our experiments show that general-purpose models often fail on subtle semantic details. In contrast, our **domain-specific IR plus validator pipeline** allows a much smaller fine-tuned model to reach **96.3% semantic correctness vs. 91.4% for GPT-4o** on the same NL→specification task. This is precisely why we care about specification flexibility and structure: it makes both human authoring and automated generation more reliable.

---

### Author Response · Authors · 2025-11-21
**Author Overall Response (1/3)**

# Author Response
We thank all reviewers for their detailed comments. We are encouraged that several reviewers view DSLs for mechanism design as a “fruitful approach” and find our system “sensible, well executed and described.” We address the main concerns below, then highlight reviewer-specific points.

---

## 1. What is actually new?

Our contribution is not just a prompt wrapper around Python, but a full stack for *verifiable* market specifications:

- **A typed market IR:** a backend-independent AST over auctions, stages, matchings, constraints, and objectives (e.g., `AuctionNode`, `ConstraintNode`). The parser only produces IR; backends only consume IR.
- **A three-stage validator pipeline on IR** (`CoreMarket`, `StageAndMatching`, `AdvancedOptimization`) that enforces syntactic, semantic, and basic economic consistency, with safe autofixes and structured reports.
- **A dual-backend execution framework** (Pandas/NetworkX simulation and CVXPY optimization) that must agree on allocations/payments for any IR, enabling cross-backend consistency checks.
- **A curated NL→DSL dataset (~11k examples)** built via a 4-stage program pipeline + description-consistency checking + human audit, and a fine-tuned 8B copilot that reaches **96.33% IR equivalence vs. 91.41%** for few-shot GPT-4o.

This architecture follows the general *“intent → copilot → DSL → compiler → backend”* pattern advocated for DSL copilots: it turns natural-language market descriptions into transparent, testable, deterministic, traceable, policy-preserving specifications rather than opaque one-off scripts.

---

## 2. IR, metrics, and “AI Copilot”

### The role of IR

The IR is the single source of truth for mechanism semantics, not just an intermediate parsing artifact. Validators, backend routing, and our evaluation metric (“IR equivalence”) are all defined over IR, not raw DSL text. IR nodes, validator report IDs, and compile artifacts are logged with each execution, so any result can be replayed with the exact spec and validator configuration. So it effectively serves as a semantic and governance layer.

We will clarify:

- **IR equivalence:** graph isomorphism between generated and reference IR ASTs, modulo variable renaming, commutative ordering, and alias normalization.
- **End-to-end correctness:** passes grammar, all three validators, compilation to both backends, IR equivalence, and execution checks on 300 scenarios.

Table 3 shows that as we move from parse-only filtering to the full IR-aware pipeline, IR equivalence jumps from **66.33%** to **96.33%** with the same 8B base model – a 30-point gain driven by IR-level validation and description–DSL alignment, not by a larger LLM.

### “AI Copilot” definition

We will define “AI Copilot” once and use it consistently to mean the **NL→DSL assistant module** (Llama-3-8B + LoRA with its prompts).  It is not the DSL itself or the whole framework, and it is distinct from the Completeness Agent, which only fills missing fields before translation.

---

## 3. Clarification on data pipeline, overfitting, and generalization

We will make the details more explicit:

- We start from **900+ use cases over 87 domains** and generate candidate (description, DSL) pairs with GPT-4o-mini, given the grammar and few-shot exemplars.
- Programs are filtered through a **4-stage pipeline**: grammar parse → three IR validators → compilation to both backends → execution without runtime errors.
- We then rewrite descriptions to be complete given the DSL and run a second LLM as a description–DSL verifier, keeping only pairs labeled “YES”.
- From the resulting pool, we form an ~**11k-example training set** and a **300-example test set**, ensuring cosine similarity < 0.85 between any train/test description (TF–IDF) and distinct IR hashes (no structural duplicates).

We agree that synthetic data may limit external validity. To demonstrate the generalization capability, we have now implemented 8 mechanisms from real ACM EC/SIGecom papers (2020–2024) (Banchio & Skrzypacz, 2022; Gui, Nair, & Niu, 2022; Feldman et al., 2022; Goke et al., 2022; Balseiro, Kroer, & Kumar, 2022; Cashore, Frazier, & Tardos, 2022; Anunrojwong, Balseiro, & Besbes, 2022). We selected mechanisms with clear, non-toy definitions. Each mechanism was encoded in MIND, validated via our IR pipeline, and executed via the appropriate backend. All 8 were expressible without changing the core grammar; we only added a small number of rule names as library entries. In the revised paper, we will add an appendix table listing these 8 mechanisms, their sources, and how they map to MIND, and summarize them briefly in Section 4.

---

### Author Response · Authors · 2025-11-21
**Author Overall Response (2/3)**

## 4. Scope, messy data, and “simple tasks”

### What MIND supports now

From the current grammar and IR (Appendix A–D), MIND can encode:

- Single- and multi-stage sealed-bid auctions with standard allocation/payment rules and reserve policies.
- Compatibility-constrained matching markets (e.g., bipartite/stable matching) with explicit compatibility graphs.
- Mechanisms with convex objectives and linear constraints that can be dispatched to the CVXPY backend (e.g., capacity/budget constraints with revenue objectives).

We do **not** claim to support fully general combinatorial auctions, arbitrary non-additive valuations, or FCC-style spectrum auctions in this version; these are explicitly listed as future work in the paper. We will add a short table summarizing “supported now” vs. “out of scope”.

### Benchmarks and complexity

We agree that the second price, a two-stage reserve auction, and a simple compatibility matching are mathematically simple. We chose them deliberately as canonical templates for three families: single-stage auctions, multi-stage reserve/re-auction processes, and compatibility-constrained matchings. They already exercise stages, cross-block coordination, and graph-based matching, while remaining easy to interpret for readers.

As suggested, we added **more complex real-world cases** as we mentioned in https://openreview.net/forum?id=DCnYQ59zrt&noteId=3ftXOuZet1 A3.

Table 1 shows that even for these “simple” mechanisms, MIND reduces implementation from **40–120 Python LoC to 10–25 DSL lines** and makes policy modifications (e.g., changing pricing rules or stages) trivial. We will make this framing explicit and add brief examples of more complex mechanisms we have encoded (e.g., multi-unit and constrained auctions from recent EC/SIGecom work) without claiming full coverage of all designs.

### Messy data and schema drift

At present, MIND assumes that auction and matching data have been normalized into tabular form (participants, goods, bids, compatibility edges) before ingestion. The DSL and IR operate *above* this layer; ETL and schema-drift handling are done with small preprocessing scripts or existing data tools. We will state this clearly and mark schema-aware extensions (e.g., detecting incompatible logs or automatically generating adapters) as orthogonal future work. This avoids silently changing market semantics when raw data formats evolve.

### NL input vs. dropdowns / GUIs

We view natural-language input as complementary to structured interfaces. Designers often start from prose policies and informal specs; our Completeness Agent and Copilot turn those into structured DSL programs. The same IR can then be driven by a form-based or dropdown UI for routine changes. We will add a sentence clarifying that MIND’s front end is pluggable: NL, forms, or both, all targeting the same DSL/IR.

---

## 5. User study and practical impact

The current draft mentions the 17-participant user study only in the abstract; we will move a concise description into Section 4.1 and an appendix. In brief:

- **Design:** preregistered within-subjects study with 17 participants, comparing mechanism modifications in MIND vs. Python/AnyLogic on the three benchmark mechanisms.
- **Tasks:** change a pricing rule, add a reserve stage, and alter compatibility constraints, starting from a working implementation in each environment.
- **Metrics:** completion time and correctness.

This directly addresses whether MIND helps with iterating on mechanisms (the common real-world task) rather than only initial implementation. We will also tie each benchmark more explicitly to real applications (e.g., reserve policies in ad auctions or government sales; compatibility matchings on platforms) and emphasize that the typed specification + validator reports + cross-backend logs form governance artifacts useful for audit and policy iteration.

---

## 6. Relation to stronger general-purpose models

Our experiments use GPT-4o as a strong proprietary baseline, showing that a smaller 8B model with our IR-aware data curation achieves higher semantic correctness (**96.33% vs. 91.41% IR equivalence**) and 100% validation/compilation success. We fully agree that newer reasoning models (e.g., GPT-5-class or Gemini-class models) are powerful; our design is front-end agnostic: any NL→DSL model can plug into the same IR, validators, backends, and governance layer.

We will clarify that our contribution is a domain-specific interface and verification pipeline that **complements**, rather than replaces, progress in general LLMs.

---

### Author Response · Authors · 2025-11-21
**Author Overall Response (3/3)**

---

## 7. Clarifications

- R1 (ia69): We will (i) emphasize IR’s central role and refine the “one engine” line into a practical backend-routing argument; (ii) define “AI Copilot” precisely as the NL→DSL assistant; and (iii) state that Table 1’s Python/AnyLogic baselines are hand-written to reflect human workflow complexity.

- R2 (yi5a): We will (i) frame our three tasks as canonical sanity checks, (ii) make the dataset pipeline, de-duplication, and audit explicit, (iii) add a clear scope table (what we support vs. out of scope), (iv) move user-study details into the main text, and (v) clarify that our main claim is about the architecture (DSL + IR + validators + dataset) rather than beating the latest closed-source models.

- R3 (N77G): We will move a motivating end-to-end example earlier (currently in Appendix G), reference Figure 1 when it appears, separate “IR equivalence” from “end-to-end correctness,” and briefly explain our use of cosine similarity for data de-duplication. We will also expand discussion of extensibility (adding rules/validators/backends) and connect to formal-methods work on auction verification as complementary tools.

- R4 (6kfJ): We will add the missing workflow-study details, a table with filtering counts at each dataset stage, a concise description of supported mechanism families, and a clearer definition of “governance artifacts” as the combination of readable DSL specs, typed IR + validator reports, and logged spec hashes/backend outputs that enable replay and audit.

We hope these clarifications address the reviewers’ concerns and better convey the novelty and practical value of MIND as a verifiable DSL copilot for market design. Below, we respond to each review separately.

---

> ### Comment · Reviewer_N77G · 2025-11-28
> **thank you for your replies to my review**
>
> Dear authors - thank you for your replies.
>
> A passing comment on comparisons to out-of-the-box LLMs, a point raised by other reviewers: my read of the field is that it now seeks to restrict the role of LLMs to orchestration whenever possible, instead supporting them with deterministic tools whenever possible.  Thus, I see this work in that light.

---

> ### Author Response · Authors · 2025-11-29
> **thank you for your replies to my review**
>
> Thank you for your reply. We strongly agree with your comment: the field is indeed moving toward using LLMs for orchestration while relying on deterministic tools for execution and verification.
>
> Our updated results in Table 2 (Section 4.2.3) reinforce your view. We found that powerful "reasoning" models like GPT-5 (thinking) and Gemini 2.5 Pro actually performed worse on end-to-end correctness (72-77%) compared to our fine-tuned model (96%), often because they hallucinate plausible but invalid syntax ("over-thinking") rather than adhering to the strict specification.
> This confirms that MIND serves exactly the role you described: it provides the deterministic, verifiable scaffolding that constrains the LLM, ensuring that the "orchestration" results in safe, executable market logic.
>
> Thank you again for championing this perspective.

---

### Note · Program_Chairs · 2026-01-17
**Submission Desk Rejected by Program Chairs**

The following references in this submission do not refer to real documents and/or have major errors in bibliographic information:

     Simon Goke, Brendan Lucier, Renato Paes Leme, and Q Wang. Bidders' responses to auction format change: Evidence from a field experiment in display ad auctions. In Proceedings of the 23rd ACM Conference on Economics and Computation, 2022.
    Hyejin Ko and Kamesh Munagala. Optimal price discrimination with a public budget. In Proceedings of the 23rd ACM Conference on Economics and Computation, pp. 943-944, 2022.